# MUONBP: FASTER MUON VIA BLOCK-PERIODIC ORTHOGONALIZATION

**Ahmed Khaled**[*]
Princeton University
ahmed.khaled@princeton.edu

**Kaan Ozkara**
AWS
kaanozka@amazon.com

**Tao Yu**
AWS
taou@amazon.com

**Mingyi Hong**
University of Minnesota, Twin Cities and AWS
mhong@umn.edu

**Youngsuk Park**
AWS
pyoungsu@amazon.com

## ABSTRACT

Gradient orthogonalization is a simple strategy that shows great utility in speeding up gradient descent. The Muon optimizer (Jordan et al., 2024b) combines gradient orthogonalization with first-order momentum and achieves significant improvement in data efficiency over Adam/AdamW (Loshchilov & Hutter, 2019a) for language model training. However, when using model parallelism, gradient orthogonalization introduces additional overhead compared to coordinate-wise optimizers (such as AdamW) due to additional gather and scatter operations on gradient matrix shards from different devices. This additional communication can amount to a throughput hit of 5%-10% compared to Adam/AdamW. To remedy this, we propose Muon with Block-Periodic Orthogonalization (MuonBP), which applies orthogonalization independently to matrix shards on each device and periodically performs full orthogonalization to maintain training stability at scale. We show how to adjust the learning rate from the baseline to MuonBP and give convergence guarantees for this algorithm. Crucially, our theory dictates that we use two stepsizes: one for the blockwise orthogonalization steps, and one for the full orthogonalization steps. Our method is simple, requires minimal hyperparameter adjustments, and achieves competitive iteration complexity compared with baseline Muon while providing per-iteration throughput comparable to coordinate-wise methods such as AdamW. When training an 8B model with eight-way tensor parallelism and ZeRO optimizer state sharding, MuonBP achieves 8% throughput increase compared to Muon with no degradation in performance.

## 1 INTRODUCTION

First order optimization methods have been the staple in the success of deep learning in the last decade. In particular, Adam (Kingma & Ba, 2015; Loshchilov & Hutter, 2019a) has become the *de facto* standard across both industry and academia. Despite numerous attempts to improve upon Adam's performance, it has remained unchallenged as the optimizer of choice for training large-scale neural networks. But this wall might be starting to crack. A recent newcomer, Muon (Jordan et al., 2024b), consistently outperforms Adam on various LLM training tasks ranging from small scale benchmarks to larger LLM training setting with up to 1T model parameters Kimi-AI et al. (2025). Muon is more data efficient than Adam, requiring fewer tokens to reach the same validation loss (Liu et al., 2025). It also enjoys a higher critical batch size, which allows for further use of parallelism (Essential AI et al., 2025) to accelerate training. Both of these aspects are critical in large-scale LLM pretraining, where even marginal efficiency gains can translate into substantial computational and financial savings.

Muon orthogonalizes the update matrix for each layer before using it in a descent step, and it can be seen as a form of steepest descent (Bernstein, 2025) or as a Non-Euclidean Trust Region

---

[*]Work done at AWS.

method (Kovalev, 2025). A key disadvantage of Muon, compared to Adam, is that orthogonalization is not a coordinate-wise operation. Rather, it requires gathering the gradient matrix from different devices whenever model parallelism is used. This introduces additional throughput overhead compared to Adam (Essential AI, 2025). Although Muon is more *token efficient*, it is strictly slower than Adam on a per-iteration basis under model parallelism.

The goal of this work is to bridge this throughput gap while preserving the data efficiency of Muon. To this end, we propose Muon with Block-Periodic orthogonalization (MuonBP, Algorithm 1). MuonBP block-orthogonalizes the matrix shards on each device independently and periodically gathers the shards for a full orthogonalization. In the off-period iterations, MuonBP does not require any additional communication, recovering the communication efficiency of Adam. However, orthogonalizing shards only is not enough for a competitive performance. We observe this block-only variant (BlockMuon, (Boreiko et al., 2025)) suffers from a potentially worse convergence guarantee and fails as the models scale up. Hence, we introduce periodic global orthogonalization steps. Combined, MuonBP recovers the performance of Muon with a drastic reduction in communication overhead. Our main contributions are as follows.

- We propose MuonBP, a variant of Muon with local orthogonalization interleaved with periodic full orthogonalization. In the off-period iterations, MuonBP treats each tensor parallel shard independently and orthogonalizes it separately. In the on-period iterations we gather the tensors and do a full orthogonalization. Our experiments with a period of 5 indicate that we recover the performance of Muon with $5\times$ reduction in the optimizer step communication volume.
- We provide a theoretical analysis of the algorithm (Theorem 2) that shows (a) the blocking period $P$ smoothly interpolates between the convergence rate of Muon and BlockMuon, that (b) we should use *two different learning rates* in the blocking vs full iterations, and finally (c) gives us guidance on how to scale the learning rate when using block orthogonalization.
- Empirically, we show that MuonBP converges faster than the baseline (non-blocking) Muon algorithm (Jordan et al., 2024b), Dion (Ahn et al., 2025), AdamW (Kingma & Ba, 2015; Loshchilov & Hutter, 2019b), and BlockMuon (Boreiko et al., 2025) in practical pretraining tasks in terms of the wall-clock time. We observe that our method recovers the original Muon's performance with a up to $8\%$ increase in throughput under layerwise sharding and tensor parallelism.

We briefly outline the rest of this paper. In Section 2, we provide necessary background for a steepest descent view of Muon, which will be useful for other sections. We discuss related work and compare our work to few others who examined orthogonalized updates in large scale distributed settings. In Section 3, we discuss our algorithm with convergence analysis, our goal is to analyze the effect of periodicity in the behaviour of our algorithm. Finally, in Section 4, we examine our algorithm in billion-scale training settings and compare to other baselines in terms of accuracy and throughput.

## 2 BACKGROUND AND RELATED WORK

Training modern large-scale machine learning models has historically relied on Adam/AdamW (Kingma & Ba, 2015; Loshchilov & Hutter, 2019a), but this might be starting to change. Recent progress on the AlgoPerf benchmark (Kasimbeg et al., 2025) and Modded-NanoGPT speedrunning (Jordan et al., 2024a) show that alternative second-order-inspired optimizers Shampoo (Gupta et al., 2018) and Muon (Jordan et al., 2024b) can be competitive or better at scale. In this section we first give a review of the algorithmic framework we use to understand it. Then, we consider the relative cost of Muon compared to Adam from a systems perspective. As we are particularly interested in communication efficiency, we also review various tools from the literature on communication-efficient optimization.

### 2.1 ALGORITHMIC FRAMEWORKS FOR OPTIMIZER ANALYSIS

The framework of steepest descent under non-Euclidean norms allows us to study different optimizers in a unified and princpled manner (Bernstein & Newhouse, 2024b). This framework is very useful in analyzing Muon as it (a) clarifies what Muon is optimizing *for*, and (b) gives a common template to compare Muon and its variants to coordinate-wise methods like Adam. Steepest descent posits that at each step of optimization, if $x$ is the current model, we choose the next model as $x + \delta x$ where $\delta x$ minimizes $f(x) + \langle \nabla f(x), x + \Delta x \rangle + \frac{\lambda}{2}\|\Delta x\|^2$, where $f$ is the loss function

and $\nabla f(x)$ is its gradient. The choice of norm $\|\cdot\|$ yields different optimizers. Choosing the Euclidean norm gives us gradient descent, and choosing the $\ell_\infty$ norm gives us scaled sign descent, $\arg\min_{\Delta x \in \mathbb{R}^d} \left( f(x) + \langle \nabla f(x), x + \Delta x \rangle + \frac{\lambda}{2}\|\Delta x\|_\infty^2 \right) = -\frac{\|\nabla f(x)\|_1}{\lambda}\mathrm{sign}(\nabla f(x))$. When exponentially moving averaging is turned off in Adam, it reduces to *unscaled* sign descent (Bernstein & Newhouse, 2024b) and there is some evidence that Adam's superior performance is explained by this connection (Kunstner et al., 2023).The steepest descent view results in the additional scaling by $\|\nabla f(x)\|_1$ in the numerator, which means we have different parameter update norm every iteration ($\propto \|\nabla f(x)\|_1$) and don't have the same connection to Adam.

We can instead explicitly control the parameter update norm by using the **Non-Euclidean Trust Region** (NTR) formulation. This is the formulation used by Kovalev (2025): at iterate $x$, NTR minimizes the first-order model of $f$ over a norm ball $\{\Delta : \| \Delta \| \le 1/\lambda\}$, which yields the steepest-descent direction in that norm. Concretely, $\Delta x = \arg\min_{\Delta \|\|\Delta\| \le \frac{1}{\lambda}} (f(x) + \langle \nabla f(x), x + \Delta x \rangle)$. For $\|\cdot\|_\infty$ this recovers (unscaled) sign descent. The NTR formulation also allows for elegant theoretical analysis, including incorporating algorithmic techniques such as momentum (Kovalev, 2025). For these reasons, we will adopt the NTR framework as our algorithmic template in Section 3.

**Muon.** Changing the norm used in either steepest descent or NTR from $\|\cdot\|_\infty$ to any other norm opens up a large design space of optimization algorithms. For example, we may use different norms for different parameters in a neural network depending on whether they are vectors or matrices. For matrix parameters $X \in \mathbb{R}^{m \times n}$, using the operator norm $\|X\|_{\mathrm{op}} = \sup_{z \in \mathbb{R}^n} \frac{\|Xz\|}{\|z\|}$ gives

$$\arg\min_{\Delta X \|\|\Delta X\|_{\mathrm{op}} \le \frac{1}{\lambda}} (f(X) + \langle \nabla f(X), X + \Delta X \rangle) = -\frac{1}{\lambda}\mathrm{Orth}(\nabla f(X)), \qquad (1)$$

where $\mathrm{Orth}(U) = (UU^\top)^{-\frac{1}{2}}U$ and $\dagger$ denotes the Moore-Penrose pseudoinverse. If we use Newton-Schulz iterations (Algorithm 2) to approximately compute the orthogonalization and apply the maximization to a running momentum buffer instead of the gradient directly, we obtain Muon (Jordan et al., 2024b). Bernstein & Newhouse (2024a) argue for using layer-dependent norms depending on the expected norm for the inputs and outputs of each layer. In practice, the choice of norm is also motivated by empirical performance (Jordan et al., 2024b). If we instead use the $\ell_1 \to \ell_2$-induced norm, we obtain column normalization. That is, given a gradient matrix $G = [G_{:,1} \quad G_{:,2} \quad \cdots \quad G_{:,n}]$, we set $\Delta X = -\frac{1}{\lambda}\left[ \frac{G_{:,1}}{\|G_{:,1}\|} \quad \cdots \quad \frac{G_{:,n}}{\|G_{:,n}\|} \right]$. This was used for the first layer in Scion (Pethick et al., 2025) and for every layer save the last in SCALE (Glentis et al., 2025). Glentis et al. (2025) show that this using column normalization with momentum on the last layer allows for training transformers competitive with Adam and Muon for up to 1B parameters scale.

## 2.2 A SYSTEMS PERSPECTIVE ON OPTIMIZER COSTS

The choice of norm dictates the operation to be done at every step and its structure (e.g. coordinate-wise vs. matrix-wise). This, in turn, determines both the computational cost of the update and whether distributed execution requires cross-device collectives.

**Computational costs.** The computational cost of running an optimizer step is just the number of floating point operations (FLOPs) we need to do per step. For methods that only perform coordinate-wise operations (such as Adam), this cost just scales with the number of parameters in the network. For methods like Muon that have to perform more sophisticated operations, this is higher. Concretely, for a parameter matrix of size $m \times n$, the per-step cost of (stochastic) gradient descent with momentum is just $2mn$ floating point operations (FLOPs) and $4mn$ FLOPs for Adam. In comparison, orthogonalization is more expensive. Using $K$ Newton-Schulz iterations (Algorithm 2 in the Appendix), the total is $2mn + 2K(2nm^2 + m^3)$ FLOPs assuming without loss of generality that $m \le n$ (Jordan et al., 2024b). Some approaches to reducing the computational cost of orthogonalization include tuning $a, b, c$ in Algorithm 2 to reduce the number of steps needed (Jordan et al., 2024b) or using adaptive per-step $a, b, c$ (Amsel et al., 2025).

In some large-scale pretraining regimes, the computational cost of running the optimizer steps might be small relative to the forward and backward passes in backpropagation. A common rule of thumb is fwd+bwd computation $\approx 6NT$ FLOPs for a dense network with $N$ params and input size of $T$ tokens. For larger batch sizes, this becomes more dominant as the optimizer step is independent of the input size.

**Communication costs.** Modern neural networks are trained with a combination of data and model parallelism. Data Parallelism (DP) replicates model parameters, gradients, and optimizer states across the communication network but passes different data batches to each DP group. The gradients are synchronized across the different devices before applying the optimizer step. While this replicates the optimizer step computation across different DP groups, it adds no additional communication cost. In contrast, model parallelism typically will shard some or all of these tensors. Tensor Parallelism (Shoeybi et al., 2019) (TP) shards the model parameters for both storage and computation; This sharding is done along one or more dimensions (e.g. row, column) of each tensor. Pipeline Parallelism (Huang et al., 2019) (PP) also shards model parameters for both storage and computation, but does so by dividing the layers among different PP groups. The Zero Redundancy Optimizer (Rajbhandari et al., 2020) (ZeRO), Fully Sharded Data Parallelism (Zhao et al., 2023) (FSDP), and FSDP2 (Liang et al., 2024) shard model parameters either by layer or on the first dimension, but do that for the purpose of saving memory. Before doing the forward/backward computation involving a certain layer, ZeRO/FSDP2 undo the sharding they apply first. In practice, we often apply a combination of parallelism strategies for maximum compute utilization, with e.g. TP applied between different devices on the same node and ZeRO/FSDP/FSDP2 applied between different nodes. Table 4 in the supplementary summarizes what the different parallelism strategies shard.

**Communication cost of Muon.** There are several strategies for parallelizing Muon and they determine the communication costs involved (Essential AI, 2025). If we use TP or FSDP2, we have to do an additional all-gather across the TP/FSDP2 groups to gather optimizer states (momentum buffers). A naive all-gather would force us to orthogonalize the same matrix in parallel which is redundant. A better alternative is to use two all-to-all communications to redistribute different layer tensors. This suffers from two issues: (a) we still have to do two additional collective operations, and (b) if the number of matrices to be orthogonalized is larger than the number of GPUs, some GPUs would sit idle. If we use ZeRO, then the fact that the optimizer states, parameters, and gradients are already sharded layerwise helps greatly: we do not need to do an all-gather across the distributed optimizer groups and can apply orthogonalization layerwise in parallel. In this case, the only extra communication cost we suffer from comes from all-gathering across the TP groups. For an 8B parameter Llama-style transformer, this gives a throughput reduction of 8%-10%

This additional communication burden has motivated the development of Dion (Ahn et al., 2025) and, concurrently to our work, Boreiko et al. (2025) introduce a variant of BlockMuon (Algorithm 1 with $P = \infty$). We compare against BlockMuon in detail in the next section and in the experiments. Dion (Ahn et al., 2025) maintains a low-rank approximation of the momentum matrix and distributes the orthogonalization process. For large enough batch sizes, Dion's computational cost is perfectly divided by the number of devices and its communication cost scales with the smaller rank. In Section C we study the computational cost of Dion in more detail and compare it to our proposed algorithm MuonBP.

**Tools for communication efficiency.** Under data parallelism, the gradient synchronization step (which happens regardless of which optimizer we use) can itself be expensive, particularly in highly distributed or federated settings (Kairouz et al., 2019). Researchers have developed techniques like gradient quantization (Alistarh et al., 2017; Horváth et al., 2019), intermittent communication (Konečný et al., 2016; Stich, 2019; Douillard et al., 2023), and low-rank compression (Vogels et al., 2019) to reduce this cost. MuLoCo (Thérien et al., 2025) applies both gradient quantization and intermittent communication to reduce the gradient synchronization cost under data parallelism, while Dion (Ahn et al., 2025) optionally allows for reducing gradient synchronization cost via low-rank compression as well. In this work, we use the same technique of intermittent communication to reduce the communication costs arising from *model* parallelism. Our algorithm can be combined with any of the aforementioned techniques for data parallelism as well.

Many of the same computational and communication constraints discussed above also apply to other gradient preconditioning algorithms, e.g. Shampoo (Gupta et al., 2018), K-FAC (Martens & Grosse, 2015), and ASGO/One-Sided Shampoo (An et al., 2025; Xie et al., 2025). Distributed Shampoo (Shi et al., 2023) uses blocking, intermittent preconditioner updates, and layer-wise sharding similar to ZeRO-1/FSDP to reduce the amount of communication.

# 3 ALGORITHMS AND CONVERGENCE

Our starting point is the observation that column- or row-wise normalization can be viewed as orthogonalization applied on a submatrix of size $m \times 1$ or $1 \times n$. An intermediate method between row-wise normalization and column-wise normalization would be orthogonalizing submatrices of dimensions $p \times q$ each where $p \leq m$ and $q \leq n$. This has two benefits,

- We reduce the amount of floating point operations per Newton-Schulz step from $2(2nm^2 + m^3)$ to $2(2pq^2 + q^3) \times \frac{mn}{pq} = 2(2mnq + \frac{mnq^2}{p})$ floating point operations (assuming without loss of generality that $p \leq q$). For example, the MLP layers in Llama 3 405B (Grattafiori et al., 2024) have $m, n \in \{53248, 16384\}$. Here, orthogonalizing submatrices with 8-way TP gives a speedup of $\approx 2.36\times$ for the up-projection and $\approx 9.06\times$ for the down-projection per Newton–Schulz step relative to full orthogonalization.
- If we use *blocks* corresponding to the model parallelism used, we can entirely eliminate orthogonalization's communication overhead under *any* regime. We discuss this in more detail below.

**How blocks align with model-parallel shards.** We divide each parameter, gradient, and optimizer state tensor into blocks and define each of these blocks to be exactly the tensor shard that resides on a device under the chosen model-parallelism layout. This makes the communication pattern explicit and ensures that a "block" step never requires cross-device traffic.

- *Tensor Parallelism (TP)*. In Megatron-style (Shoeybi et al., 2019) *column-parallel* linear layers, a weight $W \in \mathbb{R}^{m \times n}$ is split by columns across $c$ TP ranks, so each rank holds $W^{(j)} \in \mathbb{R}^{m \times (n/c)}$ and produces a local gradient shard $G^{(j)} \in \mathbb{R}^{m \times (n/c)}$. A *block* is $G^{(j)}$; block-orthogonalization acts on $m \times (n/c)$ matrices and needs no gather/scatter. In *row-parallel* layers, $W$ is split by rows across $r$ ranks, so each shard is $((m/r) \times n)$ and the block is $G^{(i)} \in \mathbb{R}^{(m/r) \times n}$. For hybrid 2D TP (row $\times$ column), the global $W$ is partitioned into an $r \times c$ grid of rectangular shards $((m/r) \times (n/c))$. TP is often applied not just to the linear layer but also to the attention weights as well, and the same discussion applies.
- *FSDP2 (dim-0 sharding)*. When parameters are sharded only for memory (layer/dim-0), each rank holds a contiguous slice along the first dimension. During the optimizer step, *block* denotes this local slice; thus block-orthogonalization again requires no parameter all-gather. The same definition applies under TP+FSDP: the block is the intersection of the TP and FSDP partitions, i.e., a single $\left(\frac{m}{r_{\text{row}}} \times \frac{n}{c_{\text{col}}}\right)$ shard.

In order to develop algorithms that minimize communication, we want to do block-wise operations as much as possible and keep "global" operations to a minimum. To this end, we analyze the variant of Muon that only does blockwise operations in Section 3.1. Our analysis shows that in the worst case, the convergence of this variant might be much worse than full Muon. To remedy this, we develop and analyze our block-periodic variant in Section 3.2.

## 3.1 BLOCK ORTHOGONALIZATION

BlockMuon (Algorithm 1 with $P = \infty$) applies orthogonalization to these blocks, in parallel, on different devices (Boreiko et al., 2025). This removes the need for any added communication and reduces the computational cost of orthogonalization. To better understand the convergence of Block-Muon, we analyze the algorithm under the assumptions of smoothness, bounded stochastic gradient variance, and norm equivalence characterized by $\rho$. We state our assumptions more clearly below.

**Assumption 1** (Smoothness). *We assume that $f : \mathbb{R}^{m \times n} \to \mathbb{R}$ is $L$-smooth with respect to a norm $\|\cdot\|$. That is, let $\|\cdot\|_*$ be the corresponding dual norm, then for all $X, Y \in \mathbb{R}^{m \times n} \to \mathbb{R}$ we assume $\|\nabla f(X) - \nabla f(Y)\|_* \leq L\|X - Y\|$.*

**Assumption 2** (Bounded Variance). *Suppose that the stochastic gradients $G(X)$ are (a) unbiased, $\mathbb{E}_\xi [G(X; \xi)] = \nabla f(X)$, and (b) have bounded variance $\mathbb{E}_\xi \left[\|G(X; \xi) - \nabla f(X)\|^2\right] \leq \sigma^2$.*

**Assumption 3** (Norm Equivalence). *The norm $\|\cdot\|$ satisfies $\|X\| \leq \rho\|X\|_{\text{F}}$ for some $\rho > 0$.*

As mentioned before, we will use the Non-Euclidean Trust Region (NTR) template (provided next) to analyze both algorithms.

$$M_t = \mu M_{t-1} + G_t, \qquad X_{t+1} = \underset{X:\|X - X_t\| \leq \eta}{\arg\min} \langle M_t, X - X_t \rangle, \qquad \text{(NTR)}$$

where $G_t$ is a stochastic gradient with expectation $\nabla f(X_t)$. This framework was adopted for the convergence analysis of Muon by Kovalev (2025) and the next theorem is a slight modification of Theorem 2 in their work. The proof is also similar to (Li & Hong, 2025, Theorem 2.1).

**Theorem 1.** *Suppose that the function $f$ satisfies Assumptions 1 to 3 and that $f$ is lower bounded by $f_*$. Then for any $\eta > 0$ and $\mu \in [0, 1]$ the iterates generated by equation NTR satisfy*

$$\mathbb{E}\left[\min_{t=0,\ldots,T-1} \|\nabla f(X_t)\|_*\right] \leq \frac{f(X_0) - f_*}{\eta T} + \frac{3\sqrt{L(f(X_0) - f_*)}}{T}\frac{\mu}{1 - \mu} \tag{2}$$
$$+ \frac{2(1-\mu)\rho\sigma}{T} + \frac{L\eta\mu}{1 - \mu} + \rho\sigma\sqrt{\frac{1 - \mu}{1 + \mu}} + \frac{L\eta}{2}.$$

Theorem 1 applies to Muon, since under $\|\cdot\| = \|\cdot\|_{\mathrm{op}}$, eq. (NTR) reduces to orthogonalizing momentum. The next lemma shows that Block-Muon can also be studied in the same framework.

**Lemma 1** (Dual of the Block-Spectral Norm). *Let $X \in \mathbb{R}^{m \times n}$ be partitioned into $r \times c$ blocks. Define the **block-spectral norm** as $B(X) = \max_{1 \leq i \leq r,\ 1 \leq j \leq c} \|X_{i,j}\|_{\mathrm{op}}$. Its dual norm is $B^*(X) = \sum_{i,j} \|X_{i,j}\|_{\mathrm{op},*}$, where $\|\cdot\|_*$ is the nuclear norm.*

BlockMuon is just eq. (NTR) with $\|\cdot\| = B(\cdot)$. To compare between the convergence of Muon and BlockMuon, we consider the simplified setting when $\sigma = 0$ and apply Theorem 1. Minimizing Equation (2) over $\eta$ and $\mu$ yields $\eta_{\mathrm{op},*} = \sqrt{\frac{2(f(X_0)-f_*)}{TL_{\mathrm{op}}}}$ and $\mu = 0$ and the convergence guarantee $\|\nabla f(X_\tau)\|_{\mathrm{op},*} \leq \sqrt{\frac{2L_{\mathrm{op}}(f(X_0)-f_*)}{T}}$, where $L_{\mathrm{op}}$ is the smoothness constant of $f$ with respect to the operator norm. Similarly, the best guarantee for BlockMuon is achieved by $\eta_{\mathrm{block},*} = \sqrt{\frac{f(X_0)-f_*}{6TL_{\mathrm{B}}}}$ and is $B^*(\nabla f(X'_\tau)) \leq \sqrt{\frac{2L_{\mathrm{B}}(f(X_0)-f_*)}{T}}$, where $\tau' = \arg\min_t B^*(\nabla f(X'_t))$ and $L_{\mathrm{B}}$ is the smoothness constant of $f$ in the block norm $B(\cdot)$. To compare the two guarantees for BlockMuon and Muon, we use the facts that $\|\cdot\|_{\mathrm{op},*} \leq B^*(\cdot)$ and $L_{\mathrm{B}} \leq rcL_{\mathrm{op}}$ (proved in Section A.1) to get $\|\nabla f(X'_{\tau'})\|_{\mathrm{op},*} \leq \sqrt{\frac{2L_{\mathrm{B}}(f(X_0)-f_*)}{T}} \leq \sqrt{rc}\sqrt{\frac{2L_{\mathrm{op}}(f(X_0)-f_*)}{T}}$. Thus, under the same operator norm metric, BlockMuon's best point $X'_{\tau'}$ has a gradient dual norm that is at most a $\sqrt{rc}$ factor worse than Muon's best point $X_\tau$ in the worst case; when $L_{\mathrm{B}} \approx L_{\mathrm{op}}$ (e.g., curvature well captured by blocks), the two bounds match up to constants. Note that in the former case, we would have $L_{\mathrm{B}} \approx (rc)L_{\mathrm{op}}$ and $\frac{\eta_{\mathrm{op},*}}{\eta_{\mathrm{block},*}} = \sqrt{\frac{L_{\mathrm{B}}}{L_{\mathrm{op}}}} = \sqrt{rc}$. Whereas, in the ideal scenario when $\eta_{\mathrm{op},*} \approx \eta_{\mathrm{block},*}$, the optimal learning rate would be the same for both algorithms. Thus *the optimal ratio of the learning rate of Block-Muon and Muon is between $1$ and $1/\sqrt{rc}$.*

The picture we see is thus clear: BlockMuon is faster on a per-step basis, as we do not need to perform any additional communication over coordinate-wise methods, but this comes at the cost of a worse convergence guarantee (by a factor of $\sqrt{rc}$ in the worst case). It seems straightforward then that we should minimize wall-clock time by choosing block sizes $r$ and $c$ that balance this tradeoff. While this is theoretically plausible, in practice the block sizes are naturally a function of network topology (i.e. FSDP or TP degrees) and changing them would add more latency and require redistributing tensors to and from their original layouts.

### 3.2 Block-periodic orthogonalization

We instead offer another alternative to tuning block sizes that (a) has a simple implementation, and (b) gives us a clear tunable knob that smoothly interpolates between BlockMuon and Muon. Given a period $P$, Muon with Block-Periodic orthogonalization instead uses BlockMuon for $\frac{P-1}{P}$ steps and then uses full orthogonalization for one step. If $P = 1$ we get Muon, while if $P \to \infty$ we get Block-Muon. Using $P$ in between both extremes allows us to balance out the tradeoff between iteration complexity and per-step communication cost. We state the algorithm in full below as Algorithm 1. Note that we use two stepsizes, $\eta_{\mathrm{full}}$ and $\eta_{\mathrm{block}}$, depending on whether we communicate during that step or not. We will later show this gives a better convergence rate than just using one stepsize.

The next theorem studies the convergence of this algorithm and allows us to make the above intuition rigorous.

**Theorem 2** (Convergence of MuonBP). *Suppose that $f$ satisfies Assumption 1 with respect to both the operator norm $\|\cdot\|_{\mathrm{op}}$ with constant $L_{\mathrm{op}}$ and the block-spectral norm $B(\cdot)$ with constant $L_{\mathrm{B}}$, and that Assumption 2 holds. Assume $f$ is lower bounded by $f_*$ and let $\Delta_0 = f(X_0) - f_*$. Fix a period $P \geq 1$, momentum $\mu \in [0, 1)$, and two stepsizes $\eta_{\mathrm{full}} > 0$ and $\eta_{\mathrm{block}} > 0$. Define $\bar{\eta} = \frac{\eta_{\mathrm{full}}}{P} + \frac{\eta_{\mathrm{block}}(P-1)}{P}$, $\eta_{\max} = \max(\eta_{\mathrm{full}}, \eta_{\mathrm{block}})$, and*

$$A = \max\{\eta_{\mathrm{full}}\sqrt{L_{\mathrm{op}}}, \eta_{\mathrm{block}}\sqrt{L_{\mathrm{B}}}\}, \qquad Q = \frac{L_{\mathrm{op}}\eta_{\mathrm{full}}^2}{2P} + \frac{L_{\mathrm{B}}\eta_{\mathrm{block}}^2(P-1)}{2P},$$

$$R = \frac{2\mu}{1-\mu}\left(\frac{L_{\mathrm{op}}\,\eta_{\mathrm{full}}\,\max\{\eta_{\mathrm{block}}\sqrt{rc}, \eta_{\mathrm{full}}\}}{P} + \frac{L_{\mathrm{B}}\,\eta_{\mathrm{block}}\,\max\{\eta_{\mathrm{full}}, \eta_{\mathrm{block}}\}(P-1)}{P}\right).$$

*Then for any horizon $T$ divisible by $P$, the iterates of Algorithm 1 satisfy*

$$\min_{t=0,\dots,T-1}\mathbb{E}\left[\|\nabla f(X_t)\|_{\mathrm{op},*}\right] \leq \frac{\Delta_0}{\bar{\eta}T} + \frac{4(1-\mu)\sigma\,\eta_{\max}}{\bar{\eta}T} + \frac{6\mu\sqrt{\Delta_0}\,A}{(1-\mu)\bar{\eta}T} + \frac{Q+R}{\bar{\eta}} + 2\sigma\sqrt{\frac{1-\mu}{1+\mu}}. \quad (3)$$

To simplify the comparison we consider the noiseless case where $\sigma = 0$ and the optimal momentum parameter is then $\mu = 0$. To minimize Equation (3), we define the harmonic-average smoothness $\bar{L}_{\mathrm{BP}}$ by $\bar{L}_{\mathrm{BP}}^{-1} = \frac{1}{P}L_{\mathrm{op}}^{-1} + \frac{P-1}{P}L_{\mathrm{B}}^{-1}$. The optimal stepsizes are then $\eta_{\mathrm{full}}^* = \frac{1}{L_{\mathrm{op}}}\sqrt{\frac{2\Delta_0}{T}\bar{L}_{\mathrm{BP}}}$ and $\eta_{\mathrm{block}}^* = \frac{1}{L_{\mathrm{B}}}\sqrt{\frac{2\Delta_0}{T}\bar{L}_{\mathrm{BP}}}$ and the convergence rate is $\min_{t<T}\|\nabla f(X_t)\|_{\mathrm{op},*} \leq \sqrt{\frac{2\Delta_0\bar{L}_{\mathrm{BP}}}{T}}$. Therefore, the convergence of BlockMuon, Muon, and MuonBP is proportional to $\sqrt{L_{\mathrm{B}}}$, $\sqrt{L_{\mathrm{op}}}$, and $\sqrt{\bar{L}_{\mathrm{BP}}}$, respectively. It is easy to see that $L_{\mathrm{op}} \leq \bar{L}_{\mathrm{BP}} \leq L_{\mathrm{B}}$ and thus the convergence rate of MuonBP is in between Muon and BlockMuon. The period $P$ acts as a tunable knob that lets us slide between the two extremes and this is directly reflected in the convergence rates we obtain. Observe that to get this rate, it is crucial that we use two stepsizes $\eta_{\mathrm{full}}$ and $\eta_{\mathrm{block}}$ depending on whether we are applying full orthogonalization or block-wise orthogonalization. On the contrary, if we were to force using a single stepsize for all steps $\eta_t \equiv \eta$, the optimal choice becomes $\eta^* = \sqrt{\frac{2\Delta_0}{T\bar{L}_{\mathrm{BP2}}}}$ with $\bar{L}_{\mathrm{BP2}} = \frac{L_{\mathrm{op}}}{P} + \frac{P-1}{P}L_{\mathrm{B}}$, yielding a convergence rate proportional to $\bar{L}_{\mathrm{BP2}}$ rather than $\bar{L}_{\mathrm{BP}}$. Since $\bar{L}_{\mathrm{BP}}$ is the weighted harmonic mean and $\bar{L}_{\mathrm{BP2}}$ is the weighted arithmetic mean of the same constants, we have $\bar{L}_{\mathrm{BP}} \leq \bar{L}_{\mathrm{BP2}}$ with strict inequality unless $L_{\mathrm{op}} = L_{\mathrm{B}}$, so tying the stepsizes generally *yields worse convergence*. Observe that, as in our previous comparison, the optimal ratio between $\eta_{\mathrm{block}}$ and $\eta_{\mathrm{full}}$ is between 1 and $1/\sqrt{rc}$.

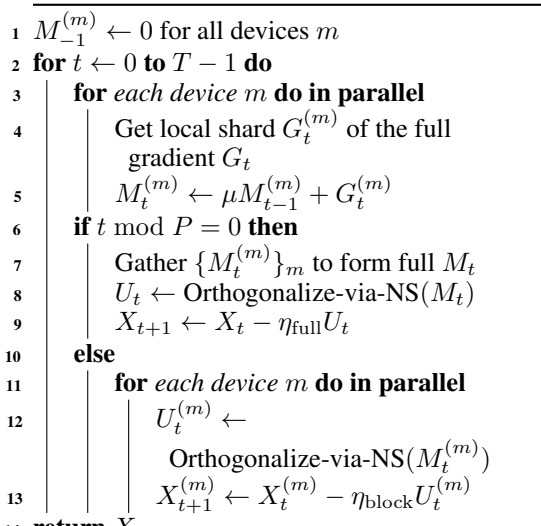

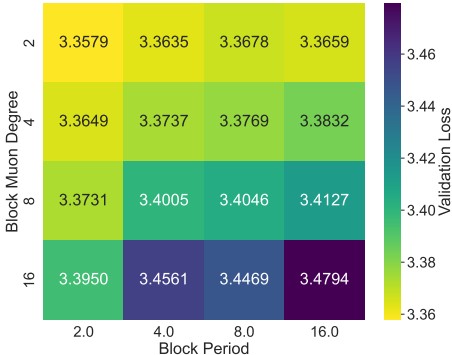

Figure 1: Validation loss as a function of orthogonalization period for different TP degrees (280M model).

**AdamW learning rate transfer.** Liu et al. (2025) introduce a learning rate scaling rule that allows reusing the AdamW learning rate for Muon by matching the root-mean square norm of the updates to be the same as AdamW. To ensure that the updates have RMS $\beta$, they scale the update matrices

by $\beta \cdot \sqrt{\max(m,n)}$ where $m \times n$ are the update matrix dimensions. Following our theorem above, which shows using different learning rates for the blocking and non-blocking matrices is ideal, we also adopt this rule and scale the updates by the dimensions of the smaller matrix on block steps and the dimensions of the full matrix on non-blocking steps.

**Communication cost of MuonBP.** On a *block* step, MuonBP orthogonolizes the local shard without optimizer-state all-gather/scatter. On a *full* step, it temporarily gathers shards to materialize $M_t$ (or $G_t$) per tensor, performs global orthogonalization, and scatters back.

**Choice of period.** Ideally $P$ minimizes the wall-clock time to reach accuracy $\varepsilon$, balancing the iterations $T_{\text{iter}}(\epsilon, P)$ needed and the expected wall-clock time per iteration $T_{\text{wall}}(P)$. This tradeoff depends on network speed and tensor size, so in practice we test $P$ on short runs. We found $P = 5$ balanced this well empirically.

## 4 EXPERIMENTS

We conduct experiments in two main settings both of which are Llama-style language model pre-training setups. Firstly, we use a setting with FSDP2 and TP where we study the effect of varying blocking degree and orthogonalization period on convergence under extensive hyperparameter tuning. Then, we benchmark our method with a small 160M model setup from (Ahn et al., 2025); and compare MuonBP to AdamW, Muon (with full all-gather at every step), BlockMuon, and Dion. FSDP2 shards optimizer states in 0th dimension to different workers, resulting in increased communication for Muon. In the second setting we use ZeRO layer-wise (Rajbhandari et al., 2020) optimizer state sharding and TP. Here, we primarily compare MuonBP (Algorithm 1), BlockMuon (Algorithm 1 with $P = \infty$), and baseline Muon (with full all-gather every step), under billion scale model sizes and longer tokens. Both experiment groups are meant to showcase the accuracy and throughput improvements brought about by our algorithm in realistic pretraining settings.

### 4.1 TRAINING WITH DIM-0 DATA SHARDING

**Experimental setting and hyperparameters.** We augment the Modded-NanoGPT codebase (Jordan et al., 2024a) with SimpleFSDP (Zhang et al., 2024) and TP (Shoeybi et al., 2019) via the DTensor API integrated into PyTorch 2.0 (Liang, 2023). We use the **FineWeb** dataset (Penedo et al., 2024) for the experiments in this section.

Figure 1 shows the effect of varying both the TP degree and the period of orthogonalization on the final validation loss achieved. We use the modernized GPT-style architecture of Modded-NanoGPT (Jordan et al., 2024a) for this experiment. We use 12 layers, 6 attention heads, and a model dimension of 768. We use the smaller model size (280M) in order to run an extensive grid search. Following the codebase, we use separate learning rates for Adam (applied to 1D parameters and the input embedding) and Muon, and do not use the RMS norm matching trick of Section 4.2. We tune the Adam/Muon learning rates over the grid $(0.0001, 0.001, 0.01, 0.1, 0.5, 1, 2, 4, 8) * \text{base}$ where $\text{base} = 0.012$ for Adam and $\text{base} = 0.08$ for Muon. We see that decreasing the block period directly decreases the loss for all the degrees we consider, with the effect most pronounced at the highest degrees.

We use the Dion codebase (Ahn et al., 2025) for the second comparison and train a 160M parameter model with a batch size of 1024, sequence length 1024, model dimension 768, 12 layers and 12 attention heads per attention layer. We use the WSD schedule with no warmup and a 20% cooldown. The learning rate is 0.02 for all methods (with AdamW rms norm matching) except for AdamW, where we found by a grid search that 0.008 performed better. We use TP degree of 2 and FSDP degree of 4, and use Lion as the scalar optimizer in line with the codebase. The throughputs for all the methods were similar at this scale, although they were significantly lower compared to throughputs on Megatron-LM with layerwise sharding. We believe more experiments are needed to compare against Dion, particularly to integrate it into widely used open source frameworks such as Megatron-LM. We also plot the loss curves in Figure 11 in Section B. Section C also gives a brief comparison of the cost of running MuonBP vs Dion from a theoretical perspective. We note that we only conducted the experimental comparison between Dion and MuonBP on a small scale, and that more work is needed to compare in this setting on a larger scale. We leave this to future work.

|                            | Muon  | BlockMuon | MuonBP | Dion  | AdamW |
| -------------------------- | ----- | --------- | ------ | ----- | ----- |
| **Min Validation Loss**    | 3.36  | 3.36      | 3.34   | 3.37  | 3.62  |
| **Min Training Loss**      | 3.02  | 2.97      | 2.94   | 2.95  | 3.21  |
| **Throughput (TFLOP/s/GPU)** | 50.90 | 51.77   | 51.40  | 45.64 | 52.80 |

Table 1: Training/validation losses and throughput on 160M model trained with TP=2 and FSDP=4.

## 4.2 TRAINING WITH LAYERWISE SHARDING

**Experimental setting and hyperparameters.** We built upon the Distributed Muon implementation of (Liu et al., 2025) in the Megatron-LM framework (Shoeybi et al., 2019) (which corresponds to ZeRO-2 optimizer state and gradient layerwise sharding) and modified it to support block-wise tensor parallel orthogonalization with periodic full orthogonalization. Note that we use the terms FSDP and ZeRO interchangeably depending on the framework, as their sharding strategies are equivalent up to minor implementation details. We used Llama-style model architecture (Touvron et al., 2023a;b) with RoPE (Su et al., 2024), SwiGLU activation (Shazeer, 2020), and mixed-precision training (bf16 computations with fp32 master weights). We use the Llama 3 tokenizer (Grattafiori et al., 2024) on the **OpenWebText** dataset (Gokaslan et al., 2019) for experiments at the 0.9-1.2B scale and the **FineWeb** data (Penedo et al., 2024) for experiments at the 8B scale. For the experiments in this section, we used nodes that have 8xA100 GPUs with 40GB of RAM each.

We train models in the following scales and settings: 960M and 1.26B, 1.26B with extended training (3x Chinchilla tokens), and 8B parameters with large ($1.2 \times 10^{-3}$) and small ($0.6 \times 10^{-3}$) learning rates. The models below 8B in scale use a batch size of 128 sequences and each run takes place on a single node with 2 DP groups and 4 TP nodes per group. The 8B model uses a batch size of 256 sequences with 4 DP groups distributed across 4 nodes and 8 TP nodes per group. As discussed in Section 3.2, we use AdamW RMS norm matching for learning rate scaling (Liu et al., 2025). All of the architectural details are provided in Table 5 in the supplementary material and more details on our choices of hyperparameters, learning rate, and learning rate scheduling are found in the appendix. We also present experiments on scheduling the period parameter $P$ in Appendix D. We do the two learning rate runs at 8B scale to show that with the larger base learning rate, even after adjusting for blocking with the RMS norm matching, BlockMuon becomes unstable.

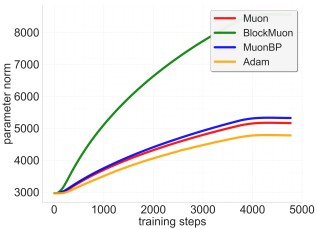

Figure 2: Parameter norm vs iteration of competing methods.

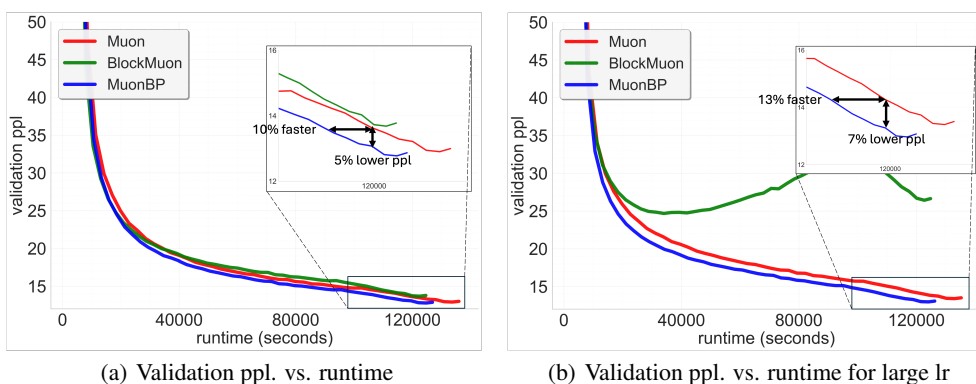

(a) Validation ppl. vs. runtime

(b) Validation ppl. vs. runtime for large lr

Figure 3: 8B model validation perplexities. Comparison of Muon, BlockMuon, and MuonBP across wall-clock time. For a target validation perplexity our method is $\sim 10 - 13\%$ faster in terms of the wall-clock time to reach it, and for a given time point before the learning rate decay our method results in $\sim 5 - 7\%$ lower perplexity compared to the baseline.

Table 2: Validation and training perplexity (*lower is better*). Columns show models; each model has validation and training sub-columns. Best perplexities within each model size are in **bold**.

| Method | 960M | | 1.2B | | 1.2B[a] | | 8B | | 8B[b] | |
|---|---|---|---|---|---|---|---|---|---|---|
| | Val | Train | Val | Train | Val | Train | Val | Train | Val | Train |
| Muon | 15.33 | 13.44 | 14.13 | 12.83 | 12.62 | 10.88 | 12.90 | 11.74 | 13.40 | 12.39 |
| BlockMuon | 20.29 | 18.08 | 16.28 | 14.86 | 13.29 | 11.51 | 13.68 | 12.62 | 24.68 | 23.17 |
| MuonBP | **15.12** | **13.21** | **13.78** | **12.44** | **12.45** | **10.71** | **12.77** | **11.59** | **12.97** | **11.93** |
| Adam | 22.51 | 20.16 | – | – | 15.03 | 13.25 | 14.47 | 13.48 | – | – |

[a] Three-times data with large learning rate (small learning rate for Adam). [b] Large learning rate.

**Results.** Resulting perplexities are summarized in Table 2. The loss curves for all models are deferred to Section B. Table 2 shows that BlockMuon performs worse in both training and validation loss across all model scales considered. This still holds true for relatively long ( 3x Chinchilla) training, as the parameter norms grow a lot more for the fully blocked version of Muon compared to either baseline or blocking with intermittent orthogonalization. Note that this happens despite the fact that we use AdamW RMS norm matching scaled with the dimensions of the sliced blocks (as outlined in Section Section 3). We observe that we have to use smaller learning rates to keep BlockMuon stable compared to Muon and MuonBP; This is potentially a symptom of the instability we observe when using BlockMuon. We do not observe instability when using smaller learning rates (Figure 10), but then baseline Muon, BlockMuon, and MuonBP all lead to the same suboptimal performance. Adam consistently underperformed across all scales, failing to converge at the larger learning rates (hence its absence in some columns). However, the performance gap between Adam and Muon also narrowed substantially with scale: the relative perplexity improvement decreased from 31.9% at 960M (22.51 vs 15.33) to 10.9% at 8B (14.47 vs 12.90), indicating that Adam's disadvantage diminishes at larger scales. This underscores the importance of MuonBP, as at larger scales it is then critical to minimize throughput losses as much as possible—even a 7% improvement in throughput would be highly significant. In Figure 3, we plot the validation ppl vs wall-clock time. We characterize our method's performance with respect to two related metric: firstly, given a target ppl value our method reaches considerably faster in wall-clock time; secondly given a runtime budget our method results in lower validation ppl (we give exemplary points in Figure 3). These two views indicate the usefulness of MuonBP in practical scenarios.

Interestingly, overall, our method outperforms Muon despite doing less number of full orthogonalization, we believe this may be due to a regularization effect due to intermittency, we leave the analysis of this behavior as future work.

**Throughput.** We report throughput numbers in Table 3. We observe similar throughput across methods in smaller scale experiments as layer-wise sharding results in minimal all-gathers for the Muon. However, as the model scale increases the effect of all-gathers makes its presence felt. Consequently, in 8B model setting we observe a $\sim 8\%$ increase in throughput for our method compared to the Muon without any degradation in performance. This translates to hundreds of thousands of dollars saved in training costs in today's large-scale pretraining runs.

Table 3: Average throughput (TFLOP/s/GPU) for each method and model.

| Method | 960M | 1.2B | 8B |
|---|---|---|---|
| Muon | 112.97 | 118.29 | 105.09 |
| BlockMuon | 115.43 | 120.14 | 114.75 |
| MuonBP | 113.54 | 119.79 | 113.37 |
| Adam | 117.21 | 120.20 | 117.30 |

## 5 CONCLUSION

We have introduced a new algorithm, MuonBP, and analyzed its convergence properties. MuonBP shows promising performance in training models up to the 8B parameter scale compared to Muon, BlockMuon, and Adam. There are many questions still left: for example, while we empirically demonstrate promising initial results for scheduling the period $P$ over the duration of training in Appendix D, how we might adaptively tune it based on observed properties remains an open question. Exploring the use of block orthogonalization with expert parallelism is also an important topic we leave to future work.

## REPRODUCIBILITY STATEMENT

Section 3 and the Appendix provide all details necessary to reproduce the theoretical results presented in this paper. Our code-base is built upon publicly available frameworks (Megatron-LM (Shoeybi et al., 2019) and Modded NanoGPT (Jordan et al., 2024a)). Section 4 and the Appendix describe the experimental settings and hyperparameters in detail. To further support reproducibility, we will release our implementation and training scripts upon publication at github.com/rka97/muonbp.

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

# Appendix

## A  MAIN PROOFS

### A.1  NORM EQUIVALENCES

**Lemma 2** (Dual of the block-spectral norm). *Let $X \in \mathbb{R}^{m \times n}$ be partitioned into $r \times c$ blocks $X_{ij} \in \mathbb{R}^{m_b \times n_b}$ (not necessarily square). Define*

$$B(X) = \max_{1 \le i \le r,\ 1 \le j \le c} \|X_{ij}\|_{\mathrm{op}}.$$

*With the Frobenius inner product $\langle X, G \rangle = \operatorname{tr}(X^\top G) = \sum_{i,j} \operatorname{tr}(X_{ij}^\top G_{ij})$, one has*

$$\sup_{B(G) \le 1} \langle X, G \rangle = \sum_{i=1}^{r} \sum_{j=1}^{c} \| X_{ij} \|_* .$$

*Moreover, if $X_{ij} = U_{ij}\Sigma_{ij}V_{ij}^\top$ is an SVD, then*

$$Z_{ij}^\star = \begin{cases} U_{ij}V_{ij}^\top, & X_{ij} \ne 0, \\ 0, & X_{ij} = 0, \end{cases}$$

*is feasible with $B(Z^\star) \le 1$ and attains the supremum:*

$$\langle X, Z^\star \rangle = \sum_{i,j} \| X_{ij} \|_* .$$

*Consequently the dual norm of $B(\cdot)$ is $B^*(Y) = \sum_{i,j} \| Y_{ij} \|_*$.*

*Proof.* For any feasible $G$ with $B(G) \le 1$, Cauchy-Schwartz gives us

$$\langle X_{ij}, G_{ij} \rangle \le \| X_{ij} \|_* \| G_{ij} \|_{\mathrm{op}} \le \| X_{ij} \|_* .$$

Summing over blocks,

$$\langle X, G \rangle = \sum_{i,j} \langle X_{ij}, G_{ij} \rangle \le \sum_{i,j} \| X_{ij} \|_* .$$

Taking the supremum over feasible $G$ yields

$$\sup_{B(G) \le 1} \langle X, G \rangle \le \sum_{i,j} \| X_{ij} \|_* .$$

We now show the above upper bound is achieved by $Z^\star$. Let $X_{ij} = U_{ij}\Sigma_{ij}V_{ij}^\top$ be an SVD and define $Z^\star$ blockwise by $Z_{ij}^\star = U_{ij}V_{ij}^\top$ if $X_{ij} \ne 0$ and $Z_{ij}^\star = 0$ otherwise. Then $\| Z_{ij}^\star \|_{\mathrm{op}} = 1$ when $X_{ij} \ne 0$ and $0$ when $X_{ij} = 0$, so $B(Z^\star) \le 1$. Moreover,

$$\langle X_{ij}, Z_{ij}^\star \rangle = \operatorname{tr}\big((U_{ij}\Sigma_{ij}V_{ij}^\top)^\top (U_{ij}V_{ij}^\top)\big) = \operatorname{tr}(\Sigma_{ij}) = \| X_{ij} \|_* .$$

Summing over blocks gives $\langle X, Z^\star \rangle = \sum_{i,j} \| X_{ij} \|_*$, which matches the upper bound, hence $Z^\star$ is optimal and the stated supremum value holds. $\qquad\square$

**Lemma 3.** *The norm equivalence constants in Assumption 3 for the operator norm and block-spectral norm are both equal to one. That is,*

$$\rho_{\mathrm{op}} = 1 \quad and \quad \rho_{\mathrm{block}} = 1.$$

*Proof.* The operator norm $\|X\|_{\mathrm{op}}$ (the largest singular value) is always less than or equal to the Frobenius norm $\|X\|_F$ (the root-sum-square of all singular values), hence $\rho_{\mathrm{op}} = 1$. The block-spectral norm is a maximum of block operator norms, $B(X) = \max_{i,j}\|X_{i,j}\|_{\mathrm{op}} \le \max_{i,j} \|X_{i,j}\|_F \le \sqrt{\sum_{i,j} \|X_{i,j}\|_F^2} = \|X\|_F$, which implies $\rho_{\mathrm{block}} = 1$ as well. $\qquad\square$

**Lemma 4.** *(Relations of norms) Suppose that $B(\cdot)$ is the block-norm corresponds to a partitioning a matrix of size $m \times n$ into $r \times c$ blocks of size $m_b \times n_b$ each. Then the following relations hold*

$$B(G) \le \|G\|_{\mathrm{op}} \le \sqrt{rc}B(G),$$

$$\|G\|_{\mathrm{op},*} \le B^*(G) \le \sqrt{rc}\|G\|_{\mathrm{op},*}.$$

*Moreover, if a function $f$ is $L_{\mathrm{op}}$-smooth with respect to the operator norm and $L_{\mathrm{B}}$-smooth with respect to the block-norm, we have*

$$L_{\mathrm{op}} \le L_{\mathrm{B}} \le (rc) \cdot L_{\mathrm{op}}.$$

*Proof.* Write $G$ as an $r \times c$ block matrix with blocks $G_{ij} \in \mathbb{R}^{m_b \times n_b}$ and recall $B(G) = \max_{i,j}\|G_{ij}\|_{\mathrm{op}}$. We first prove that $B(G) \le \|G\|_{\mathrm{op}}$. Let $R_i \in {0,1}^{m_b \times m}$ select the $i$-th block of rows and $C_j \in {0,1}^{n \times n_b}$ select the $j$-th block of columns, so $G_{ij} = R_i G C_j$. Because $R_i$ and $C_j$ are partial isometries, $\|R_i\|_{\mathrm{op}} = \|C_j\|_{\mathrm{op}} = 1$. By submultiplicativity,

$$\|G_{ij}\|_{\mathrm{op}} = \|R_i G C_j\|_{\mathrm{op}} \le \|R_i\|_{\mathrm{op}}\|G\|_{\mathrm{op}}\|C_j\|_{\mathrm{op}} = \|G\|_{\mathrm{op}}.$$

Taking the maximum over $(i,j)$ yields $B(G) \le \|G\|_{\mathrm{op}}$.

Now we prove the upper bound on $\|G\|_{\mathrm{op}}$. Let $u \in \mathbb{R}^m$, $v \in \mathbb{R}^n$ be unit vectors and partition them as $u = [u_1 \quad \dots \quad u_r]$, $v = [v_1 \quad \dots \quad v_c]$ with $u_i \in \mathbb{R}^{m_b}$ and $v_j \in \mathbb{R}^{n_b}$. Then

$$\begin{aligned}
|u^\top G v| &= \Big| \sum_{i=1}^r \sum_{j=1}^c u_i^\top G_{ij} v_j \Big| \\
&\le \sum_{i=1}^r \sum_{j=1}^c |u_i^\top G_{i,j} v_j| \\
&\le \sum_{i=1}^r \sum_{j=1}^c \|u_i\|_2 \|G_{ij}\|_{\mathrm{op}} \|v_j\|_2 \\
&\le B(G) \left( \sum_{i=1}^r \|u_i\|_2 \right) \left( \sum_{j=1}^c \|v_j\|_2 \right) \\
&\le B(G)\sqrt{r}\sqrt{\sum_{i=1}^r \|u_i\|_2^2}\sqrt{c}\sqrt{\sum_{j=1}^c \|v_j\|_2^2} \\
&= \sqrt{rc}B(G),
\end{aligned}$$

where we used Cauchy–Schwarz applied to the vectors of blockwise $\ell_2$ norms. Taking the supremum over unit $u, v$ gives $\|G\|_{\mathrm{op}} \le \sqrt{rc}B(G)$.

For the dual norm bounds, observe that if norms $||\cdot||_a$ and $||\cdot||_b$ satisfy $\alpha||\cdot||_a \le ||\cdot||_b \le \beta||\cdot||_a$, then their duals satisfy

$$\frac{1}{\beta}||\cdot||_a^* \le ||\cdot||_b^* \le \frac{1}{\alpha}||\cdot||_a^*.$$

Applying this with $||\cdot||_a = B(\cdot), ||\cdot||_b = \|\cdot\|_{\mathrm{op}}, \alpha = 1, \beta = \sqrt{rc}$ yields

$$\|G\|_{\mathrm{op},*} \le B^*(G) \le \sqrt{rc}\|G\|_{\mathrm{op},*}.$$

Finally, for the smoothness bounds, we have for any $X \ne Y$

$$\begin{aligned}
\frac{\|\nabla f(X) - \nabla f(Y)\|_{\mathrm{op},*}}{\|X - Y\|_{\mathrm{op}}} &\le \frac{B^*(\nabla f(X) - \nabla f(Y))}{\|X - Y\|_{\mathrm{op}}} \\
&\le \frac{B^*(\nabla f(X) - \nabla f(Y))}{B(X - Y)} \le \frac{\sqrt{rc}\|\nabla f(X) - \nabla f(Y)\|_{\mathrm{op},*}}{B(X - Y)} \\
&\le (rc)\frac{\|\nabla f(X) - \nabla f(Y)\|_{\mathrm{op},*}}{\|X - Y\|_{\mathrm{op}}}.
\end{aligned}$$

Taking the supremum over $X, Y$ such that $X \ne Y$ immediately yields $L_{\mathrm{op}} \le L_{\mathrm{B}} \le (rc)L_{\mathrm{op}}$. $\quad\square$

## A.2 CONVERGENCE PROOFS

**Lemma 5** (Descent Lemma). *Suppose that Assumptions 1 and 2 hold for $f : \mathbb{R}^{m \times n} \to \mathbb{R}$. Then the iterations of the algorithm without a regularizer satisfy:*

$$f(X_{k+1}) \leq f(X_k) - \eta \|\nabla f(X_k)\| + 2\eta \|\nabla f(X_k) - (1 - \mu) M_k\|_* + \frac{3}{2} L \eta^2.$$

*Proof.* By smoothness,

$$f(X_{k+1}) \leq f(X_k) + \langle \nabla f(X_k), X_{k+1} - X_k \rangle + \frac{L}{2} \|X_{k+1} - X_k\|_*^2$$

$$\leq f(X_k) - \eta \langle \nabla f(X_k), \mathrm{Orth}(M_k) \rangle + \frac{L\eta^2}{2}$$

$$= f(X_k) - \eta \langle \nabla f(X_k) - (1 - \mu) M_k, \mathrm{Orth}(M_k) \rangle - \eta(1 - \mu) \langle M_k, \mathrm{Orth}(M_k) \rangle + \frac{L\eta^2}{2}$$

$$\leq f(X_k) + \eta \|\nabla f(X_k) - (1 - \mu) M_k\|_* \|\mathrm{Orth}(M_k)\| - \eta(1 - \mu) \|M_k\|_* + \frac{L\eta^2}{2}$$

$$\leq f(X_k) + \eta \|\nabla f(X_k) - (1 - \mu) M_k\|_* - \eta \|(1 - \mu) M_k\|_* + \frac{L\eta^2}{2}$$

$$\leq f(X_k) + 2\eta \|\nabla f(X_k) - (1 - \mu) M_k\|_* - \eta \|\nabla f(X_k)\|_* + \frac{L\eta^2}{2}.$$

$\square$

**Lemma 6.** *Suppose that $f$ satisfies Assumption 1 in some norm $\|\cdot\|$ and is lower bounded by $f_*$, then*

$$\|\nabla f(X)\|_*^2 \leq 2L \left( f(X) - f_* \right).$$

*Proof.* Define

$$X_+ = \underset{Y}{\arg \min} \left[ f(X) + \langle \nabla f(X), Y - X \rangle + \frac{L}{2} \|Y - X\| \right] = X - \eta \|\nabla f(X)\|_* Z,$$

where $Z$ satisfies $\|Z\| \leq 1$ and $\langle \nabla f(X), Z \rangle = \|\nabla f(X)\|_*$. By smoothness,

$$f(X_+) \leq f(X) + \langle \nabla f(X), X_+ - X \rangle + \frac{L}{2} \|X_+ - X\|^2$$

$$= f(X) - \eta \|\nabla f(X)\|_*^2 + \frac{L\eta^2}{2} \|\nabla f(X)\|_*^2$$

$$= f(X) - \eta \left( 1 - \frac{L\eta}{2} \right) \|\nabla f(X)\|_*^2.$$

Plugging $\eta = \frac{1}{L}$ gives

$$f_* \leq f(X_+) \leq f(X) - \frac{\|\nabla f(X)\|_*^2}{2L}.$$

Therefore,

$$\|\nabla f(X)\|_*^2 \leq 2L \left( f(X) - f_* \right).$$

$\square$

**Lemma 7.** *Let $f$ satisfy Assumptions 1 to 3 in some norm $\|\cdot\|$ with smoothness constant $L$, stochastic gradient variance $\sigma^2$, and $\ell_2$-norm ratio $\rho$. Let $M_\tau$ be defined as*

$$M_\tau = \mu M_{\tau-1} + G_\tau,$$
$$X_{\tau+1} = X_\tau - \eta_\tau Z_\tau,$$

*where $\|Z_\tau\| \eta_\tau \leq A$ for all $\tau \leq k$. Then,*

$$\mathbb{E} \left[ \|\nabla f(X_k) - (1 - \mu) M_k\|_* \right] \leq \mu^k (1 - \mu)^2 \rho \sigma + \mu^{k+1} \sqrt{2L\Delta_0} + \frac{LA\mu}{1 - \mu} + \rho \sigma \sqrt{\frac{1 - \mu}{1 + \mu}}.$$

*Proof.* Let $M_k' = (1-\mu)M_k$. We have,

$$
\begin{aligned}
\nabla f(X_k) - M_k' &= \nabla f(X_k) - \left[\mu M_{k-1}' + (1-\mu)G_k\right] \\
&= \nabla f(X_k) - \mu M_{k-1}' - (1-\mu)G_k \\
&= \nabla f(X_k) - G_k - \mu M_{k-1}' + \mu G_k \\
&= \nabla f(X_k) - G_k + \mu \nabla f(X_{k-1}) - \mu M_{k-1}' + \mu G_k - \mu \nabla f(X_{k-1}) \\
&= \nabla f(X_k) - G_k + \mu \nabla f(X_{k-1}) - \mu M_{k-1}' + \mu G_k - \mu \nabla f(X_k) + \mu \nabla f(X_k) - \mu \nabla f(X_{k-1}) \\
&= (1-\mu)\left(\nabla f(X_k) - G_k\right) + \mu(\nabla f(X_{k-1}) - M_{k-1}') + \mu\left(\nabla f(X_k) - \nabla f(X_{k-1})\right).
\end{aligned}
$$

Let $E_k = \nabla f(X_k) - M_k'$, $S_k = \nabla f(X_k) - G_k$, and $R_k = \nabla f(X_k) - \nabla f(X_{k-1})$. Then the above is,

$$
\begin{aligned}
E_k &= \mu E_{k-1} + (1-\mu)S_k + \mu R_k \\
&= \mu^k E_0 + \sum_{j=0}^{k-1} \mu^j \left[(1-\mu)S_{k-j} + \mu R_{k-j}\right].
\end{aligned}
$$

Now observe that

$$
\|R_{k-j}\|_* = \|\nabla f(X_{k-j}) - \nabla f(X_{k-j-1})\|_* \le L\|X_k - X_{k-j-1}\| = L\eta_k\|Z_k\| \le LA.
$$

Therefore

$$
\begin{aligned}
\mathbb{E}\left[\|E_k\|_*\right] &\le \mu^k \mathbb{E}\left[\|E_0\|_*\right] + \mathbb{E}\left[\left\|\sum_{j=0}^{k-1}\mu^j[(1-\mu)S_{k-j} + \mu R_{k-j}]\right\|_*\right] \\
&\le \mu^k \mathbb{E}\left[\|E_0\|_*\right] + \sum_{j=0}^{k-1}\mu^{j+1}\mathbb{E}\left[\|R_{k-j}\|_*\right] + \mathbb{E}\left[\left\|\sum_{j=0}^{k-1}\mu^j(1-\mu)S_{k-j}\right\|_*\right] \\
&\le \mu^k \mathbb{E}\left[\|E_0\|_*\right] + LA\sum_{j=0}^{k-1}\mu^{j+1} + \rho\mathbb{E}\left[\left\|\sum_{j=0}^{k-1}\mu^j(1-\mu)S_{k-j}\right\|_2\right] \\
&\le \mu^k \mathbb{E}\left[\|E_0\|_*\right] + \frac{LA\mu}{1-\mu} + \rho\sqrt{\mathbb{E}\left[\left\|\sum_{j=0}^{k-1}\mu^j(1-\mu)S_{k-j}\right\|_2^2\right]}
\end{aligned}
\tag{4}
$$

We have

$$
\begin{aligned}
\mathbb{E}\left[\left\|\sum_{j=0}^{k-1}\mu^j(1-\mu)S_{k-j}\right\|_2^2\right] &= \mathbb{E}\left[\sum_{j=0}^{k-1}\sum_{i=0}^{k-1}\mu^j(1-\mu)^2\mu^i\left\langle S_{k-j}, S_{k-i}\right\rangle\right] \\
&= \sum_{j=0}^{k-1}\mu^{2j}(1-\mu)^2\mathbb{E}\left[\|S_{k-j}\|^2\right] \\
&\le \sigma^2(1-\mu)^2\sum_{j=0}^{k-1}\mu^{2j} \\
&\le \frac{\sigma^2(1-\mu)^2}{1-\mu^2} \\
&= \frac{\sigma^2(1-\mu)^2}{(1-\mu)(1+\mu)} \\
&= \frac{\sigma^2(1-\mu)}{1+\mu}.
\end{aligned}
\tag{5}
$$

Using eq. (5) back in eq. (4) we get

$$\mathbb{E}\left[\|E_k\|_*\right] \le \mu^k \mathbb{E}\left[\|E_0\|_*\right] + \frac{LA\mu}{1-\mu} + \rho\sigma\sqrt{\frac{1-\mu}{1+\mu}}. \tag{6}$$

For the first term,

$$
\begin{aligned}
\mathbb{E}\left[\|E_0\|_*\right] &= \mathbb{E}\left[\|\nabla f(X_0) - (1-\mu)M_0\|_*\right] \\
&= \mathbb{E}\left[\|\nabla f(X_0) - (1-\mu)G_0\|_*\right] \\
&= \mathbb{E}\left[\|(1-\mu)(\nabla f(X_0) - G_0) + \mu\nabla f(X_0)\|_*\right] \\
&\le (1-\mu)\mathbb{E}\left[\|\nabla f(X_0) - G_0\|_*\right] + \mu\|\nabla f(X_0)\|_*.
\end{aligned}
$$

By Lemma 6 we have $\|\nabla f(X_0)\|_* \le \sqrt{2L\left(f(X_0) - f_*\right)}$ and by Assumptions 2 and 3 we have $\mathbb{E}\left[\|\nabla f(X_0) - G_0\|_*\right] \le \rho\sigma$. Plugging this back in gives

$$\mathbb{E}\left[\|E_0\|_*\right] \le (1-\mu)\rho\sigma + \mu\sqrt{2L\Delta_0},$$

where $\Delta_0 = f(X_0) - f_*$. Using the last equation in Equation (6) yields

$$\mathbb{E}\left[\|E_k\|_*\right] \le \mu^k(1-\mu)^2\rho\sigma + \mu^{k+1}\sqrt{2L\Delta_0} + \frac{LA\mu}{1-\mu} + \rho\sigma\sqrt{\frac{1-\mu}{1+\mu}}.$$

$\square$

*Proof of Theorem 1.* By Lemmas 5 and 7 we have

$$
\begin{aligned}
\mathbb{E}\left[f(X_{k+1})\right] &\le \mathbb{E}\left[f(X_k)\right] - \eta\mathbb{E}\left[\|\nabla f(X_k)\|_*\right] + 2\eta\mathbb{E}\left[\|\nabla f(X_k) - (1-\mu)M_k\|_*\right] + \frac{L\eta^2}{2} \\
&\le \mathbb{E}\left[f(X_k)\right] - \eta\mathbb{E}\left[\|\nabla f(X_k)\|_*\right] + 2\eta\mu^k(1-\mu)^2\rho\sigma + 2\eta\mu^{k+1}\sqrt{2L\Delta_0} \\
&\quad + \frac{2L\eta^2\mu}{1-\mu} + 2\eta\rho\sigma\sqrt{\frac{1-\mu}{1+\mu}} + \frac{L\eta^2}{2}.
\end{aligned}
$$

Rearranging,

$$
\begin{aligned}
\mathbb{E}\left[\|\nabla f(X_k)\|_*\right] &\le \frac{1}{\eta}\left[\mathbb{E}\left[f(X_k)\right] - \mathbb{E}\left[f(X_{k+1})\right]\right] + 2\mu^k(1-\mu)^2\rho\sigma + 2\mu^{k+1}\sqrt{2L\Delta_0} \\
&\quad + \frac{L\eta\mu}{1-\mu} + \rho\sigma\sqrt{\frac{1-\mu}{1+\mu}} + \frac{L\eta^2}{2}.
\end{aligned}
$$

Summing up both sides as $k = 0, 1, \ldots, T-1$ and telescoping

$$
\begin{aligned}
\sum_{k=0}^{T-1}\mathbb{E}\left[\|\nabla f(X_k)\|_*\right] &\le \frac{1}{\eta}\left[f(X_0) - \mathbb{E}\left[f(X_T)\right]\right] + 2(1-\mu)^2\rho\sigma\sum_{k=0}^{T-1}\mu^k + 2\mu\sqrt{2L\Delta_0}\sum_{k=0}^{T-1}\mu^k \\
&\quad + \frac{L\eta\mu T}{1-\mu} + \rho\sigma T\sqrt{\frac{1-\mu}{1+\mu}} + \frac{L\eta}{2}. \\
&\le \frac{\Delta_0}{\eta} + 2(1-\mu)\rho\sigma + \frac{2\mu\sqrt{2L\Delta_0}}{1-\mu} + \frac{L\eta\mu T}{1-\mu} + \rho\sigma T\sqrt{\frac{1-\mu}{1+\mu}} + \frac{L\eta}{2}.
\end{aligned}
$$

Dividing both sides by $T$ and lower bounding the average on the left hand side by the minimum yields the theorem's statement. $\square$

*Proof of Theorem 2.* Let Assumption 3 hold for both norms with constants $\rho_{\mathrm{op}}$ and $\rho_{\mathrm{block}}$ respectively and let $\rho_{\mathrm{BP}} = \frac{\rho_{\mathrm{op}}}{P} + \frac{P-1}{P}\rho_{\mathrm{block}}$. We will later show that $\rho_{\mathrm{op}}, \rho_{\mathrm{block}}$, and $\rho_{\mathrm{BP}}$ are all bounded by 1. Let $k \le T-1$. If $k$ is divisible by $P$, then by Lemma 5 we have

$$
\begin{aligned}
\mathbb{E}\left[f(X_{k+1})\right] &\le \mathbb{E}\left[f(X_k)\right] - \eta_{\mathrm{full}}\mathbb{E}\left[\|\nabla f(X_k)\|_{\mathrm{op},*}\right] \\
&\quad + 2\eta_{\mathrm{full}}\mathbb{E}\left[\|\nabla f(X_k) - (1-\mu)M_k\|_{\mathrm{op},*}\right] + \frac{L_{\mathrm{op}}\eta_{\mathrm{full}}^2}{2}.
\end{aligned}
\tag{7}
$$

To apply Lemma 7 with the operator norm, we need to ensure that the updates $Z_\tau = \frac{X_\tau - X_{\tau-1}}{\eta}$ always satisfy $\|Z_\tau\|_{\mathrm{op}}\eta_\tau \leq A$ for all $\tau \leq k$, where $\eta_\tau$ is the stepsize used on the $\tau$-th iteration. Observe that on all $\tau$ such that $\tau\%P = 0$, this is trivially true with $A = \eta_{\mathrm{full}}$. On steps where $\tau$ is not divisible by $P$, we have $\|Z_\tau\|_{\mathrm{op}} \leq \sqrt{rc}B(Z_\tau) \leq \sqrt{rc}$ (see Section A.1), since on those steps $\eta_\tau = \eta_{\mathrm{block}}$ we therefore have $\|Z_\tau\|_{\mathrm{op}}\eta_\tau \leq \eta_{\mathrm{block}}\sqrt{rc}$. Therefore in all cases, $\|Z_\tau\|_{\mathrm{op}}\eta_\tau \leq \max(\eta_{\mathrm{block}}\sqrt{rc}, \eta_{\mathrm{full}})$ and we can apply Lemma 7 to get

$$\mathbb{E}\left[\|\nabla f(X_k) - (1-\mu)M_k\|_{\mathrm{op},*}\right] \leq \mu^k(1-\mu)^2\rho_{\mathrm{op}}\sigma + \mu^{k+1}\sqrt{2L_{\mathrm{op}}\Delta_0}$$
$$+ \frac{L_{\mathrm{op}}\mu}{1-\mu}\max(\eta_{\mathrm{full}}, \eta_{\mathrm{block}}\sqrt{rc}) + \rho_{\mathrm{op}}\sigma\sqrt{\frac{1-\mu}{1+\mu}}.$$

We can then use this to bound the third term on the right hand side of Equation (7) to get

$$\mathbb{E}\left[f(X_{k+1})\right] \leq \mathbb{E}\left[f(X_k)\right] - \eta_{\mathrm{full}}\mathbb{E}\left[\|\nabla f(X_k)\|_{\mathrm{op},*}\right] + 2\eta_{\mathrm{full}}\mu^k(1-\mu)^2\rho_{\mathrm{op}}\sigma$$
$$+ 2\eta_{\mathrm{full}}\mu^{k+1}\sqrt{2L_{\mathrm{op}}\Delta_0} + \frac{2L_{\mathrm{op}}\eta^2\sqrt{rc}\mu}{1-\mu} + 2\eta\rho_{\mathrm{op}}\sigma\sqrt{\frac{1-\mu}{1+\mu}} + \frac{L_{\mathrm{op}}\eta^2}{2}. \quad (8)$$

Alternatively, if $k$ is not divisible by $P$, then similar to the above we first use Lemma 5 and the fact that $B^*(\nabla f(X_k)) \geq \|\nabla f(X_k)\|_{\mathrm{op},*}$ to get

$$\mathbb{E}\left[f(X_{k+1})\right] \leq \mathbb{E}\left[f(X_k)\right] - \eta_{\mathrm{block}}\mathbb{E}\left[\|\nabla f(X_k)\|_{\mathrm{op},*}\right] + 2\eta_{\mathrm{block}}\mathbb{E}\left[B^*(\nabla f(X_k) - (1-\mu)M_k)\right]$$
$$+ \frac{L_{\mathrm{B}}\eta_{\mathrm{block}}^2}{2}. \quad (9)$$

We now apply Lemma 7 to the block norm. Note that if $\tau$ is not divisible by $P$, $B(Z_\tau)\eta_\tau = B(Z_\tau)\eta_{\mathrm{block}} \leq \eta_{\mathrm{block}}$. If $\tau$ is divisible by $P$, then $B(Z_\tau)\eta_\tau = B(Z_\tau)\eta_{\mathrm{full}} \leq \|Z_\tau\|_{\mathrm{op}}\eta_{\mathrm{full}} \leq \eta_{\mathrm{full}}$. Therefore by Lemma 7 we have

$$\mathbb{E}\left[B^*(\nabla f(X_k) - (1-\mu)M_k)\right] \leq \mu^k(1-\mu)^2\rho_{\mathrm{block}}\sigma + \mu^{k+1}\sqrt{2L_{\mathrm{B}}\Delta_0} + \frac{L_{\mathrm{B}}\max(\eta_{\mathrm{block}}, \eta_{\mathrm{full}})\mu}{1-\mu}$$
$$+ \rho_{\mathrm{block}}\sigma\sqrt{\frac{1-\mu}{1+\mu}}. \quad (10)$$

Using Equation (10) in Equation (9) we obtain

$$\mathbb{E}\left[f(X_{k+1})\right] \leq \mathbb{E}\left[f(X_k)\right] - \eta_{\mathrm{block}}\mathbb{E}\left[\|\nabla f(X_k)\|_{\mathrm{op},*}\right] + \frac{L_{\mathrm{B}}\eta_{\mathrm{block}}^2}{2}$$
$$+ 2\eta_{\mathrm{block}}\left[\mu^k(1-\mu)^2\rho_{\mathrm{block}}\sigma + \mu^{k+1}\sqrt{2L_{\mathrm{B}}\Delta_0} + \frac{L_{\mathrm{B}}\max(\eta_{\mathrm{block}}, \eta_{\mathrm{full}})\mu}{1-\mu} + \rho_{\mathrm{block}}\sigma\sqrt{\frac{1-\mu}{1+\mu}}\right]. \quad (11)$$

Let $S_P = \{k < T \mid k \pmod P = 0\}$ and $S_B = \{k < T \mid k \pmod P \neq 0\}$. By rearranging the one-step descent inequalities and using Equations (8) and (11) we sum over $k = 0, \ldots, T-1$:

$$\sum_{k=0}^{T-1}(\mathbf{1}_{k\in S_P}\eta_{\mathrm{full}} + \mathbf{1}_{k\in S_B}\eta_{\mathrm{block}})\mathbb{E}\left[\|\nabla f(X_k)\|_{\mathrm{op},*}\right] \leq \sum_{k=0}^{T-1}(\mathbb{E}\left[f(X_k)\right] - \mathbb{E}\left[f(X_{k+1})\right])$$
$$+ \sum_{k\in S_P}(\mathrm{Error}_k^{\mathrm{full}}) + \sum_{k\in S_B}(\mathrm{Error}_k^{\mathrm{block}}), \quad (12)$$

where

$$\mathrm{Error}_k^{\mathrm{full}} = 2\eta_{\mathrm{full}}\mu^k(1-\mu)^2\rho_{\mathrm{op}}\sigma + 2\eta_{\mathrm{full}}\mu^{k+1}\sqrt{2L_{\mathrm{op}}\Delta_0} + \frac{2L_{\mathrm{op}}\eta_{\mathrm{full}}\max(\eta_{\mathrm{block}}\sqrt{rc}, \eta_{\mathrm{full}})\mu}{1-\mu}$$
$$+ 2\eta_{\mathrm{full}}\rho_{\mathrm{op}}\sigma\sqrt{\frac{1-\mu}{1+\mu}} + \frac{L_{\mathrm{op}}\eta_{\mathrm{full}}^2}{2},$$

$$\mathrm{Error}_k^{\mathrm{block}} = 2\eta_{\mathrm{block}}\mu^k(1-\mu)^2\rho_{\mathrm{block}}\sigma + 2\eta_{\mathrm{block}}\mu^{k+1}\sqrt{2L_{\mathrm{B}}\Delta_0} + \frac{2L_{\mathrm{B}}\eta_{\mathrm{block}}\max(\eta_{\mathrm{block}}, \eta_{\mathrm{full}})\mu}{1-\mu}$$
$$+ 2\eta_{\mathrm{block}}\rho_{\mathrm{block}}\sigma\sqrt{\frac{1-\mu}{1+\mu}} + \frac{L_{\mathrm{B}}\eta_{\mathrm{block}}^2}{2}.$$

The first term on the right hand side of eq. (12) is a telescoping sum bounded by $\Delta_0 = f(X_0) - f_*$. For the error terms dependent on $\mu^k$, we can form a simple upper bound by summing over all $k$ and using the larger constants $(L_B, \rho_{\text{block}})$:

Momentum-dependent error

$$= \sum_{k \in S_P} \left[ 2\eta_{\text{full}} \mu^k (1-\mu)^2 \rho_{\text{op}} \sigma + 2\eta_{\text{full}} \mu^{k+1} \sqrt{2L_{\text{op}} \Delta_0} \right]$$

$$+ \sum_{k \in S_B} \left[ 2\eta_{\text{block}} \mu^k (1-\mu)^2 \rho_{\text{block}} \sigma + 2\eta_{\text{block}} \mu^{k+1} \sqrt{2L_B \Delta_0} \right]$$

$$\leq 4 \sum_{k=0}^{T-1} \left( \mu^k (1-\mu)^2 \max(\eta_{\text{full}} \rho_{\text{op}}, \eta_{\text{block}} \rho_{\text{block}}) \sigma + \mu^{k+1} \max(\eta_{\text{full}} \sqrt{L_{\text{op}}}, \eta_{\text{block}} \sqrt{L_B}) \sqrt{2L_B \Delta_0} \right)$$

$$\leq 4 \left( (1-\mu) \max(\eta_{\text{full}} \rho_{\text{op}}, \eta_{\text{block}} \rho_{\text{block}}) \sigma + \frac{\mu \max(\eta_{\text{full}} \sqrt{L_{\text{op}}}, \eta_{\text{block}} \sqrt{L_B}) \sqrt{2\Delta_0}}{1-\mu} \right).$$

For the other terms, we sum them proportionally. Observe that $|S_P| = T/P$ and $|S_B| = T(P-1)/P$, as we assume that $T$ is divisible by $P$. Therefore,

$$\text{Constant error} = \sum_{k \in S_P} \left( \frac{L_{\text{op}} \eta_{\text{full}}^2}{2} + \frac{2L_{\text{op}} \max(\eta_{\text{full}} \eta_{\text{block}} \sqrt{rc}, \eta_{\text{full}}^2) \mu}{1-\mu} + 2\eta_{\text{full}} \rho_{\text{op}} \sigma \sqrt{\frac{1-\mu}{1+\mu}} \right)$$

$$+ \sum_{k \in S_B} \left( \frac{L_B \eta_{\text{block}}^2}{2} + \frac{2L_B \max(\eta_{\text{full}} \eta_{\text{block}}, \eta_{\text{block}}^2) \mu}{1-\mu} + 2\eta_{\text{block}} \rho_{\text{block}} \sigma \sqrt{\frac{1-\mu}{1+\mu}} \right)$$

$$= T \left( \frac{\eta_{\text{full}}^2 L_{\text{op}}}{2P} + \frac{L_B \eta_{\text{block}}^2 (P-1)}{2P} + \right.$$

$$+ \frac{2\mu}{1-\mu} \left[ \frac{L_{\text{op}} \max(\eta_{\text{full}} \eta_{\text{block}} \sqrt{rc}, \eta_{\text{full}}^2)}{P} + \frac{L_B \max(\eta_{\text{full}} \eta_{\text{block}}, \eta_{\text{block}}^2)(P-1)}{P} \right]$$

$$\left. + 2\sigma \sqrt{\frac{1-\mu}{1+\mu}} \left[ \frac{\eta_{\text{full}} \rho_{\text{op}}}{P} + \frac{\eta_{\text{block}} \rho_{\text{block}} (P-1)}{P} \right] \right)$$

Observe that,

$$\sum_{k=0}^{T-1} (\mathbb{1}_{k \in S_P} \eta_{\text{full}} + \mathbb{1}_{k \in S_B} \eta_{\text{block}}) \mathbb{E} \left[ \|\nabla f(X_k)\|_{\text{op},*} \right] \geq T \left[ \frac{\eta_{\text{full}}}{P} + \frac{\eta_{\text{block}} (P-1)}{P} \right] \min_k \mathbb{E} \left[ \|\nabla f(X_k)\|_{\text{op},*} \right]$$

$$= T\bar{\eta} \min_k \mathbb{E} \left[ \|\nabla f(X_k)\|_{\text{op},*} \right].$$

Observe that for the operator norm and the block spectrum norm, we have $\rho_{\text{op}} \leq 1$ and $\rho_{\text{block}} \leq 1$ by Lemma 3. Using this and combining all the error parts we get

$$\bar{\eta} T \min_k \mathbb{E} \left[ \|\nabla f(X_k)\|_{\text{op},*} \right] \leq \Delta_0 + 4(1-\mu)\sigma \eta_{\max} + \frac{6\mu \sqrt{\Delta_0}}{1-\mu} \max(\eta_{\text{full}} \sqrt{L_{\text{op}}}, \eta_{\text{block}} \sqrt{L_B})$$

$$+ T \left( \frac{\eta_{\text{full}}^2 L_{\text{op}}}{2P} + \frac{L_B \eta_{\text{block}}^2 (P-1)}{2P} + 2\sigma \sqrt{\frac{1-\mu}{1+\mu}} \bar{\eta} \right.$$

$$\left. + \frac{2\mu}{1-\mu} \left[ \frac{L_{\text{op}} \eta_{\text{full}} \max(\eta_{\text{block}} \sqrt{rc}, \eta_{\text{full}})}{P} + \frac{L_B \eta_{\text{block}} \max(\eta_{\text{full}}, \eta_{\text{block}})(P-1)}{P} \right] \right).$$

Dividing both sides by $\bar{\eta} T$ yields

$$\min_{k < T} \mathbb{E} \left[ \|\nabla f(X_k)\|_{\text{op},*} \right] \leq \frac{\Delta_0}{\bar{\eta} T} + \frac{4(1-\mu)\sigma \eta_{\max}}{\bar{\eta} T} + \frac{6\mu \sqrt{\Delta_0}}{1-\mu} \cdot \frac{\max\{\eta_{\text{full}} \sqrt{L_{\text{op}}}, \eta_{\text{block}} \sqrt{L_B}\}}{\bar{\eta} T}$$

$$+ \frac{1}{\bar{\eta}} \left[ \frac{L_{\text{op}} \eta_{\text{full}}^2}{2P} + \frac{L_B \eta_{\text{block}}^2 (P-1)}{2P} \right.$$

$$\left. + \frac{2\mu}{1-\mu} \left( \frac{L_{\text{op}} \eta_{\text{full}} \max\{\eta_{\text{block}} \sqrt{rc}, \eta_{\text{full}}\}}{P} + \frac{L_B \eta_{\text{block}} \max\{\eta_{\text{full}}, \eta_{\text{block}}\}(P-1)}{P} \right) \right] + 2\sigma \sqrt{\frac{1-\mu}{1+\mu}},$$

| Approach | Sharded (P/G/O) | How sharded | Params unsharded in F/B? |
|---|---|---|---|
| No parallelism | None | — | No |
| Tensor Parallelism | P+G+O | (any dim) | No |
| FSDP2 (dim-0) | P+G+O | dim-0 | Yes |
| ZeRO-1 | O | layer | Yes |
| ZeRO-2 | G+O | layer | Yes |
| FSDP/ZeRO-3 | P+G+O | layer | Yes |
| PP | P+G+O | layer | No |

Table 4: "Sharded" lists which of parameters (P), gradients (G), and optimizer states (O) are partitioned across ranks. F/B = forward/backward. TP=Tensor Parallelism. PP=Pipeline Parallelism.

where

$$\bar{\eta} = \frac{\eta_{\text{full}}}{P} + \frac{P-1}{P}\,\eta_{\text{block}}, \qquad \eta_{\max} = \max\{\eta_{\text{full}}, \eta_{\text{block}}\}.$$

$\square$

# B  ADDITIONAL ALGORITHMS, RESULTS, AND EXPERIMENTAL DETAILS

A summary of different parallelism strategies is given in Table 4. The orthogonalization via Newton Schulz iterations procedure is stated as Algorithm 2.

---

**Algorithm 2:** Orthogonalize-via-NS($G$, $K$, $\varepsilon = 10^{-7}$)

1   $a \leftarrow 2,\ b \leftarrow -1.5,\ c \leftarrow 0.5$;
2   $X \leftarrow G/(\|G\| + \varepsilon)$;
3   **for** $t \leftarrow 1$ **to** $K$ **do**
4      $A \leftarrow XX^{\top}$;
5      $B \leftarrow bA + cA^2$;
6      $X \leftarrow aX + BX$;
7   **return** $X$;

---

All the architectural hyperparameters and learning rates used are given in Table 5 below. We used the learning rates from (Liu et al., 2025) as a starting point for each run. We found at a smaller scale that the learning rates recommended therein could be increased by a factor of $\approx 3$ with no harm to convergence for the baseline (Muon) algorithm, and we scaled the other learning rates accordingly. Nevertheless, we do two experiments with smaller learning rates. We use GQA (Ainslie et al., 2023), RoPE (Su et al., 2024), bf16 mixed-precision training, and a weight decay value of 0.1. We also apply gradient clipping with value 1.0 to the parameters optimized by AdamW (mainly 1D parameters and the input embedding). We use cosine decay with no warmup for the 960M and 1.2B experiments and the Warmup-Stable-Decay (WSD) schedule (Hägele et al., 2024) with linear decay to $4.2 \times 10^{-5}$ for the 8B model.

Table 5: Section 4.2 experiments hyperparameters (sequence length = 8K, Batch size=128 sequences for 960M/1.2B models and 256 sequences for 8B models).

| Model | Layers | Heads | Query Groups | Hidden Size | (DP, TP) | LR ($\times 10^{-3}$) | Tokens (B) | Steps |
|---|---|---|---|---|---|---|---|---|
| 960M | 12 | 16 | 4 | 1536 | (2, 4) | 3.503 | 9.503 | 9063 |
| 1.2B | 14 | 16 | 4 | 1792 | (2, 4) | 3.291 | 14.143 | 13488 |
| 1.2B (3x long, larger lr) | 14 | 16 | 4 | 1792 | (2, 4) | 3.291 | 42.143 | 40191 |
| 1.2B (3x long, smaller lr) | 14 | 16 | 4 | 1792 | (2, 4) | 0.86 | 42.143 | 40191 |
| 8B (smaller lr) | 32 | 32 | 8 | 4096 | (4, 8) | 0.6 | 9.99 | 4764 |
| 8B (larger lr) | 32 | 32 | 8 | 4096 | (4, 8) | 1.2 | 9.99 | 4764 |

We report the training curves for the 960M model in Figure 4, for the 1.2B model in Figure 5, for the 1.2B model trained to 3x Chinchilla with smaller learning rate in Figure 7, with larger learning

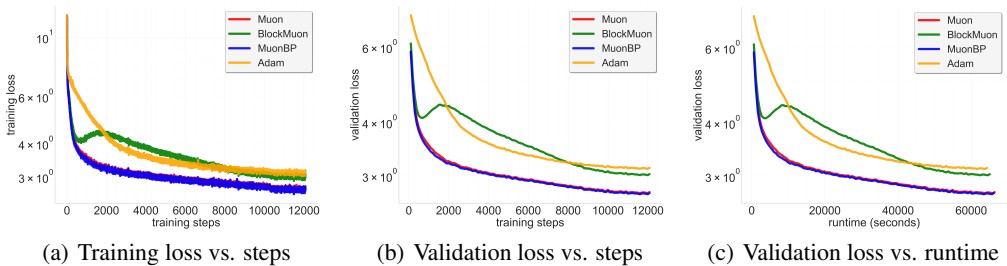

(a) Training loss vs. steps      (b) Validation loss vs. steps      (c) Validation loss vs. runtime

Figure 4: 960M model. Comparison of baseline, block, and periodic orthogonal block methods across training steps and wall-clock time.

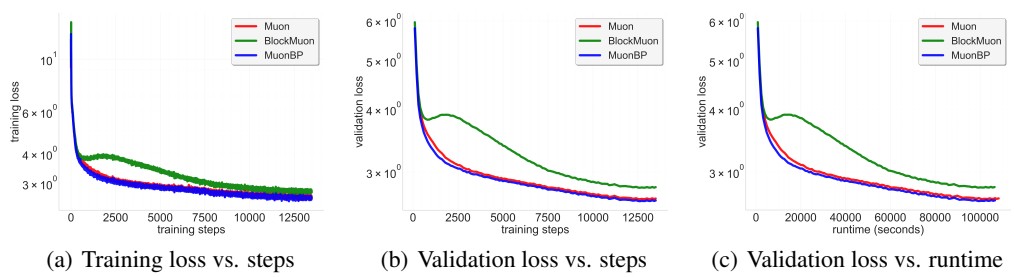

(a) Training loss vs. steps      (b) Validation loss vs. steps      (c) Validation loss vs. runtime

Figure 5: 1.2B model. Comparison of baseline, block, and periodic orthogonal block methods across training steps and wall-clock time.

in Figure 6, and for the 8B model in Figure 10 (smaller learning rate) and Figure 9 (larger learning rate). Our main observation here is that even after doing RMS norm adjustment (i.e. update is scaled by $\sqrt{\max(A, B)}$ where $A$ and $B$ are the dimensions of the update matrix, which scale inversely with blocking), BlockMuon can become unstable with larger learning rates. On the contrary, MuonBP does not.

A reason why this might be the case is that BlockMuon almost always causes the parameter norms to grow larger over time compared to Muon or MuonBP, as can be seen in Table 6. This holds even when we use small learning rates and learning rate adjustment.

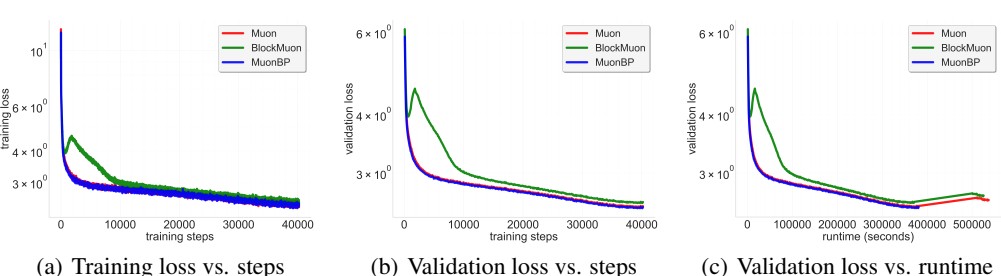

(a) Training loss vs. steps      (b) Validation loss vs. steps      (c) Validation loss vs. runtime

Figure 6: 1.2B model (larger lr), trained to 3x Chinchilla. Comparison of baseline, block, and periodic orthogonal block methods across training steps and wall-clock time.

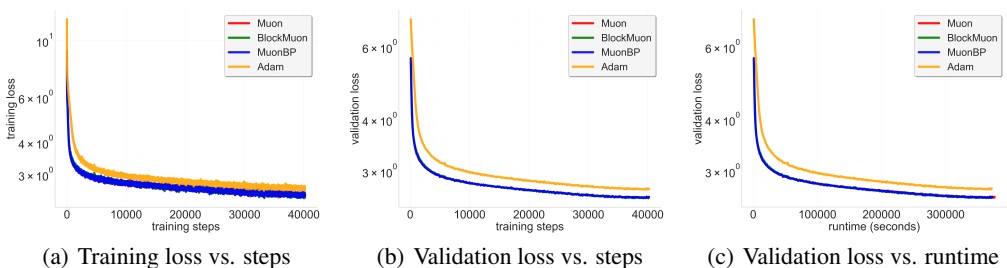

(a) Training loss vs. steps     (b) Validation loss vs. steps     (c) Validation loss vs. runtime

Figure 7: 1.2B model (smaller lr), trained to 3x Chinchilla. Comparison of baseline, block, and periodic orthogonal block methods across training steps and wall-clock time.

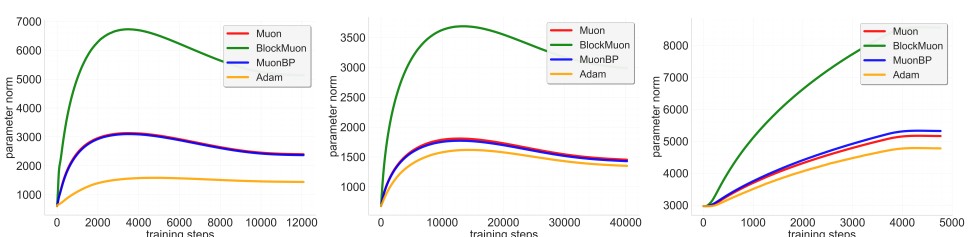

Figure 8: Comparison of parameter norms using Muon, BlockMuon, and MuonBP over training on 960M model (left), 1.2B model 3x-Chinchilla with smaller lr (center), and 8B model (right).

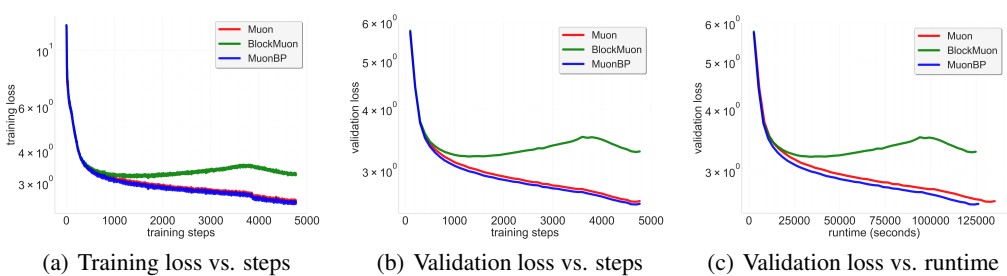

(a) Training loss vs. steps     (b) Validation loss vs. steps     (c) Validation loss vs. runtime

Figure 9: 8B model (larger lr). Comparison of baseline, block, and periodic orthogonal block methods across training steps and wall-clock time.

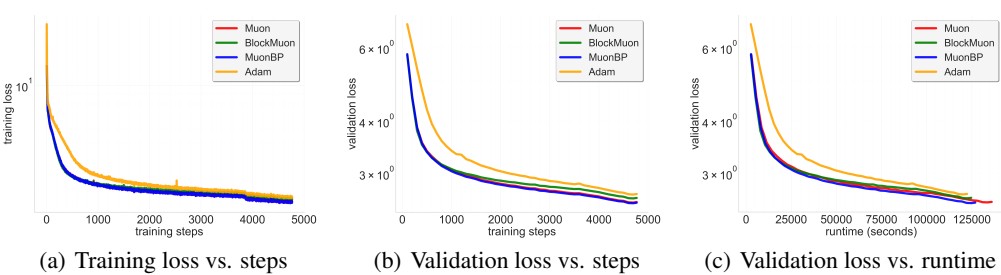

(a) Training loss vs. steps     (b) Validation loss vs. steps     (c) Validation loss vs. runtime

Figure 10: 8B model (smaller lr). Comparison of baseline, block, and periodic orthogonal block methods across training steps and wall-clock time.

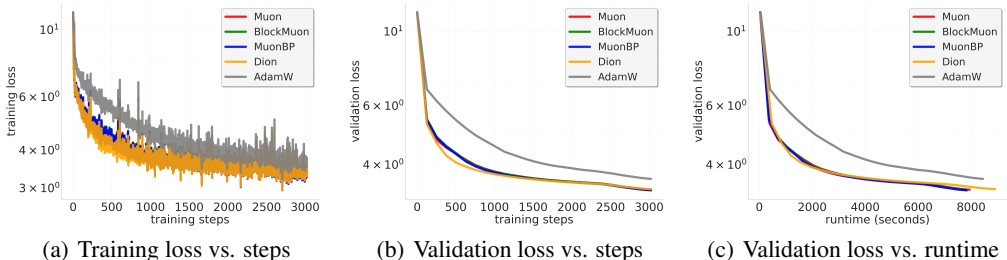

(a) Training loss vs. steps     (b) Validation loss vs. steps     (c) Validation loss vs. runtime

Figure 11: 160M model training with 2-way FSDP2 and 4-way TP. Comparison of Baseline, Block-Muon, MuonBP, and Dion across training steps and wall-clock time.

Table 6: Validation/Training perplexity (*lower is better*) and average parameter norm. Best perplexities within each model size are in **bold**.

| Model | Method | Val PPL | Train PPL | Param Norm |
|---|---|---|---|---|
| 960M | Muon | 15.34 | 13.43 | 2680 |
| | BlockMuon | 20.28 | 18.09 | 5702 |
| | MuonBP | **15.11** | **13.21** | 2648 |
| | Adam | 22.51 | 20.16 | 1433 |
| 1.2B | Muon | 14.12 | 12.83 | 4237 |
| | BlockMuon | 16.29 | 14.86 | 7225 |
| | MuonBP | 13.78 | 12.44 | 4195 |
| 1.2B (3×, large lr) | Muon | 12.62 | 10.88 | 2681 |
| | BlockMuon | 13.29 | 11.51 | 5521 |
| | MuonBP | **12.45** | **10.71** | 2868 |
| 1.2B (3×, small lr) | Muon | 13.27 | 11.40 | 1602 |
| | BlockMuon | 13.23 | 11.29 | 3242 |
| | MuonBP | 13.30 | 11.39 | 1571 |
| | Adam | 15.03 | 13.25 | 1448 |
| 8B | Muon | 12.90 | 11.74 | 4369 |
| | BlockMuon | 13.68 | 12.62 | 6680 |
| | MuonBP | **12.77** | **11.59** | 4471 |
| | Adam | 14.47 | 13.48 | 4104 |
| 8B (large lr) | Muon | 13.40 | 12.39 | 6841 |
| | BlockMuon | 24.68 | 23.17 | 11 496 |
| | MuonBP | 12.97 | 11.93 | 7063 |

## C  COMPARISON AGAINST DION

**Memory usage.**  For a weight matrix $X \in \mathbb{R}^{m \times n}$, DION maintains the usual momentum buffer $M \in \mathbb{R}^{m \times n}$ together with a right subspace basis $V \in \mathbb{R}^{n \times r}$ (of rank $r$), yielding persistent state memory usage $O(mn + nr)$. Some transient memory is allocated to "skinny" factor matrices (e.g., $U \in \mathbb{R}^{m \times r}$, $W \in \mathbb{R}^{n \times r}$) and small $r \times r$ solves, totaling $O(mr + nr + r^2)$ for transient memory usage. MUONBP keeps only the momentum state per shard (globally $O(mn)$), i.e., no persistent low-rank bases. However, on the periodic "full" step it temporarily materializes the full tensor to orthogonalize globally before scattering back, which introduces an $O(mn)$ transient buffer on that step; the intermediate "block" steps operate locally on shards and require only shard-sized temporaries.

**Compute per iteration.**  Unsharded DION performs an amortized low-rank power-iteration and orthonormalization whose leading cost scales as $\mathcal{O}(mnr + mr^2 + r^3 + mn)$ per update. In contrast, MUONBP uses Newton–Schulz (NS) orthogonalization (Algorithm 2). On blocking steps, MuonBP runs NS on a local $p \times q$ shard with cost proportional to that block (per NS iteration), while every $P$ steps it executes one global NS on the full matrix (cubic in the larger dimension). Averaged over a period, the per-iteration cost is $\frac{P-1}{P}$ times the block cost plus $\frac{1}{P}$ of a full NS. This comes out to

$$\mathcal{O}\left( \frac{P-1}{P}\left(pq^2 + q^3\right) \ + \ \frac{1}{P}\left(mn^2 + n^3\right) \right).$$

**Communication per iteration.**  With model parallelism, DION communicates only low-rank objects each step, $O(mr)$ bits along the axis that forms or broadcasts $U$ and $O(nr)$ bits along the axis that forms $W$, plus $O(r^2)$ micro-collectives for orthonormalization; critically, nothing scales with $mn$. Under data parallel, Dion can synchronize its low-rank optimizer state and gradients, reducing DP I/O from $O(mn)$ to $O((m + n)r)$. For MUONBP, block steps incur no optimizer-specific model-parallel traffic beyond the baseline training collectives, while the periodic full step performs a gather/orthogonalize/scatter of the entire tensor (layout dependent), so the average optimizer communication volume $O(mn/P)$. We can see that $m/P$ or $n/P$ act as the counterpart of Dion's rank $r$ parameter in MuonBP. For both algorithms, we need to tune one parameter (the rank $r$ vs the communication period $P$).

## D  ADAPTIVE AND SCHEDULED PERIOD $P$

While MuonBP uses a fixed period $P$ for periodic orthogonalization in all of our main experiments, we also explored adaptively scheduling $P$ to further reduce communication overhead without sacrificing convergence quality. We hypothesize that as training progresses, we can afford less frequent global orthogonalizations.

To test this, we conducted an ablation study using the 960M parameter model architecture described in Section 4.2. We trained the model for 2.37 billion tokens over 3000 iterations with a batch size of 96 sequences and a tensor parallelism degree of 8. We compared four different configurations:

   (a) **Muon (Baseline)**: Full orthogonalization at every step ($P = 1$).
   (b) **BlockMuon**: Fully blocked orthogonalization with no global synchronization ($P = \infty$).
   (c) **MuonBP (Scheduled $P$)**: The period $P$ is increased linearly from 2 to 20 over the course of training.
   (d) **MuonBP (Fixed Expected $P$)**: The period is fixed at $P = 11$, which is approximately the expected value of the linear schedule.

Our experimental results, shown in Figure 12 and Table 7, indicate that MuonBP with the scheduled $P$ ($2 \rightarrow 20$) achieves a final validation perplexity of 18.01. This outperforms both the baseline Muon, which achieves 18.28, and the fully blocked BlockMuon, which degrades to 28.60 validation perplexity. Moreover, MuonBP with a fixed expected period of $P = 11$ achieved a slightly higher validation perplexity of 18.11 compared to the scheduled variant. Notably, the scheduled MuonBP reached the baseline Muon's final perplexity around step 2800 ($\sim 6\%$ faster in wall-clock time compared to baseline Muon). These results demonstrate that scheduling $P$ can offer a promising

avenue for accelerating convergence, and we believe further exploration of adaptive or scheduled periods is an interesting direction for future work.

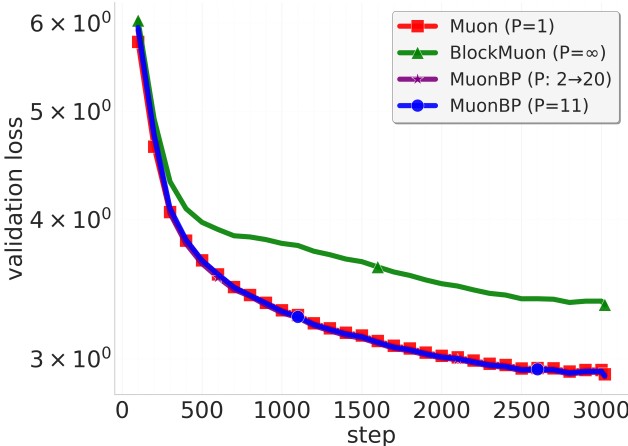

Figure 12: Validation loss across training steps for different scheduling regimes for the period $P$.

Table 7: Final Validation Perplexity for Period Scheduling Experiments.

| Method | Validation Perplexity |
|---|---|
| Muon (Baseline, $P = 1$) | 18.28 |
| BlockMuon ($P = \infty$) | 28.60 |
| MuonBP (Scheduled $P : 2 \rightarrow 20$) | **18.01** |
| MuonBP (Fixed $P = 11$) | 18.11 |

