# OpenReview forum: "MuonBP: Faster Muon via Block-Periodic Orthogonalization"
_ICLR.cc/2026/Conference — ICLR 2026 Poster_

### Official Review · Reviewer_Wrom · 2025-10-28

**Soundness:** 2
**Presentation:** 2
**Contribution:** 2
**Rating:** 2
**Confidence:** 3

**Summary:**

This paper proposed MuonBP, which basically trying to solve BlockMuon's loss divergence issue by adding some global synchronization via extra communication and adjust learning rate accordingly. the end results shows MuonBP loss curve getting lower than BlockMuon.

**Strengths:**

1.adding some global sync on top of BlockMuon to make loss curve converge better is a contribution.

2.PPL analysis is detailed and reasonable.

**Weaknesses:**

1. Research novelty is minor,

2. Comparing loss(PPL) with running time is misleading. By looking at figure 3, it is hard to see that baseline Muon trains less iteration thus loss curve is higher than proposed MuonBP. If the author want to show the iteration time, they can create a table on this. In model training, the most important thing is model quality, which means after training same amount of Iterations (Not same amount of time), which model converges to the lowest PPL or loss values. The second priority is iteration time.

3. The whole paper does not even include how many iterations each training is in their experiments, which is quite concerning.

4. The authors lack of basic distributed training knowledge, for example L147-148, " If we use TP or FSDP2, we have to
do an additional all-gather across the TP/FSDP2 groups to gather the model parameters", which is Wrong. TP never gather model parameters.

**Questions:**

The whole paper uses fsdp and zero interchangeably, which makes it quite confusing. What zero stage is indeed used? zero 1,2,3. (please describe this more important and delete some unnecessary details like row-wise split, which is the by-default setting)

Also sometime the experiment says using Megatron? how could megatron-lm used together with simpleFSDP? they don't have the synchronization implemented and tested between TP and this DP.

---

> ### Author Response · Authors · 2025-11-20
>
> Thank you for your review, please find our responses below.
>
> - **On research novelty.** We respectfully disagree. Our contributions include solving a critical problem for large-scale optimization: Muon suffers 5-10% throughput loss under model parallelism, BlockMuon attempts to fix this but fails at larger scales (as shown by Table 2, achieving a perplexity of 24.68 vs 13.4 for the baseline) and our paper provides a solution to this problem. While intermittent communication has been used for gradient accumulation, its application to orthogonalization instead is novel and indeed, non-trivial. It is not clear apriori why this should work at all; Our theory provides an explanation for why this is the case and also shows how to set the learning rates appropriately. None of this has been shown in any prior literature to the best of our knowledge.
> - **On comparing loss vs runtime**.  Figure 3 shows that MuonBP reaches **the same target perplexity** faster. Table 2 compares final model quality **for the same number of tokens** and clearly shows MuonBP also achieves the lowest perplexities. Appendix Figures 4-11 show the loss versus the number of steps, and MuonBP consistently achieves the lowest perplexity values. ““If the author want to show the iteration time, they can create a table on this.” We did indeed do that as well, the purpose of Table 3 is to show the **throughput** for each algorithm at different scales. The throughput is directly related to the iteration time, as the number of floating point operations is the same for all the methods in the table. Tables 2 and 3 combined show that our algorithm achieves lower perplexity with a faster iteration speed.
> - **On iteration counts.** We presume that ‘iteration’ mentioned by the reviewer corresponds to the number of optimization steps. Table 4 (in the Appendix) specifies the batch size, sequence length, and total number of tokens for each experiment. From this information, it is trivial to calculate the number of steps as the total number of tokens / (batch size x sequence length). All the Appendix figures (Figures 4-11) have the step counts on the x-axis. It is common in many papers to report the total number of tokens processed (e.g. as in the Chinchilla scaling laws paper) rather than the number of iterations, but we will also add the number of steps to Table 4 in the revised version.
> - **For TP and parameter gathering.** We gather **optimizer states (momentum buffers)**, not model parameters. **Algorithm 1, Line 7** clearly shows: "Gather ${M\_t^(m)}\_m$ to form full $M\_t$" where $M\_t$ is the momentum buffer. Our implementation in Megatron-LM performs all-gather on momentum states across TP groups before orthogonalization, and this is consistent with the Distributed Muon baseline of Liu et al. (2025). Saying we do an all-gather on model parameters in Lines 147-148 is a typo and we apologize for the confusion. We will correct it in the revision.
> - **On FSDP/ZeRO terminology**. We use the Distributed Optimizer implementation in the Megatron-LM experiments (Section 4.2) which corresponds to **ZeRO-2** (optimizer state \+ gradient layerwise sharding) in communication cost (see https://docs.nvidia.com/megatron-core/developer-guide/latest/api-guide/dist\_optimizer.html). Our code follows the Distributed Muon baseline of Liu et al. (2025), “Muon is Scalable for LLM Pretraining”. We will clarify this further in the revision. We use **FSDP** interchangeably with **ZeRO** since the PyTorch implementation correspond to ZeRO up to relatively minor differences, see https://[huggingface.co/docs/accelerate/en/usage\_guides/fsdp\#mapping-between-fsdp-sharding-strategies-and-deepspeed-zero-stages](http://huggingface.co/docs/accelerate/en/usage_guides/fsdp#mapping-between-fsdp-sharding-strategies-and-deepspeed-zero-stages); Note the granularities in both frameworks, both frameworks have 3 options corresponding to each other. We will make this more clear in the revision. We do not perform any experiments with PyTorch’s FSDP, but instead use SimpleFSDP/FSDP2, both of which correspond to dimension-0 sharding (as mentioned in the paper).

---

> ### Author Response · Authors · 2025-11-20
>
> - On experimental settings: We conduct experiments in **two distinct settings**:
>   1. **Section 4.1**: SimpleFSDP (Zhang et al., 2024\) \+ TP on Modded-NanoGPT codebase (160M, 280M models)
>   2. **Section 4.2**: Megatron-LM with ZeRO-2 \+ TP (960M, 1.2B, 8B models)
>
>   These are separate codebases. This is stated in:
>   - Section 4.1: "We augment the Modded-NanoGPT codebase with SimpleFSDP...".
>   - Section 4.2: "We built upon the Distributed Muon implementation... in the Megatron-LM framework".
>   - Section headings explicitly distinguish the two settings.
>   Therefore, we do not use Megatron-LM with FSDP2 or SimpleFSDP.
>
> Given the apparent confusions, we sincerely hope that our responses above have clarified these confusions. It’d be great if you could revisit our paper (especially Section 4\) and the experimental setup descriptions in the appendix. We’re happy to address any further questions.

---

### Official Review · Reviewer_aDQM · 2025-11-01

**Soundness:** 4
**Presentation:** 3
**Contribution:** 3
**Rating:** 8
**Confidence:** 5

**Summary:**

This paper proposes MuonBP, a communication-efficient variant of the Muon optimizer that combines block-wise orthogonalization with periodic full orthogonalization. It is a good attempt to scale Muon to modern training setup, e.g., model parallelism. The authors provide theoretical convergence guarantees, showing that MuonBP interpolates between BlockMuon (fully local) and Muon (fully global), delivering 8% throughput increase compared to Muon with no degradation in performance.

**Strengths:**

* **Clear motivation and practical importance.** The work directly addresses a key limitation of Muon, its poor scalability under model parallelism, by reducing communication overhead. It is a big issue for Muon as I know.
* **Theoretical grounding** The authors rigorously gives a bound for the propose method.
* **Realistic pretraining setting evaluation**. Experiments on multiple setups (FSDP2 + TP, ZeRO + TP, and scales from 160M to 8B parameters) demonstrate that MuonBP recovers Muon’s data efficiency with significantly better wall-clock performance.
* **Clear and good writing**: the paper is clear, well-motivated, and easy-to-follow.

**Weaknesses:**

Overall, it is a good paper that solves a real problem. I only have a few questions:
* **Exploration beyond pretraining**. All experiments focus on pretraining, I am curious whether the author can envision the performance on SFT or even RL work.
* **How to adaptively tune the P**: The Author mentions that we might adaptively tune it based on observed properties. I wonder whether the author has some insight on how to monitor, especially in the practical training setting with multi-paralelesim

**Questions:**

n/a

---

> ### Author Response · Authors · 2025-11-20
>
> Thank you for your review. We appreciate your detailed evaluation and comments on the clarity, motivation, and results of our work.
> - **On exploration beyond pretraining**. For supervised fine-tuning (SFT), we expect different optimizers to show smaller differences than in pretraining, as SFT typically uses much less computation and shorter training horizons. The throughput penalty is most visible at larger scales in pre-training. That said, MuonBP's lower per-step overhead compared to Muon means it remains a practical choice. We expect the same advantage to transfer from pre-training to SFT, at least if the pretraining optimizer is the same, as Liu et al. (2025) in “Muon is scalable for LLM training” observed that SFT with Muon on models pretrained with Adam/AdamW results in worse performance than on models trained with Muon. Unfortunately due to time & resource constraints we could not run additional SFT or RL experiments.  We agree that a systematic evaluation of MuonBP on SFT and RL is warranted and we leave it to future work.
> - **Adaptively tuning P**. This is a difficult problem, but we do have a signal that we can already monitor in distributed settings (the parameter norms), which we observe are particularly high for BlockMuon and in line with baseline Muon for MuonBP with a reasonable period (Figures 2 and 8 and Table 5 in the appendix). The main issue is how to get the baseline value for this signal (i.e. the expected parameter norms). \[You can check the history parameter norms, e.g. compare with the exponential averaged threshold, isn’t it?\] One way would be to instead track the derivative of the parameter norms and decrease P when this derivative exceeds a certain threshold. According to our theoretical analysis, the optimal P also depends on the ratio between the smoothness constant of the block norm and the operator norm, but estimating these smoothness constants without additional function/gradient evaluations is not easy. We have not figured out the right way to do this yet.
> - **Scheduling P instead.** We did perform additional experiments on scheduling P instead. While this is not adaptive, it can also serve to reduce the amount of communication without sacrificing quality. Using the same 960M parameter model as in Section 4.2, we ran training for 2.37 billion tokens over 3000 iterations with batch size 96 and TP degree 8\. A schedule that increases P linearly from 2 to 20 over training performed well: (a) BlockMuon (P=∞): 28.60 validation perplexity (155.5 minutes), (b) Muon (baseline): 18.28 validation perplexity (154.3 minutes), (c) MuonBP (scheduled P: 2→20): 18.01 final validation perplexity (155.6 minutes). Notably, MuonBP reached Muon's final perplexity at step 2800 (\~145 minutes), achieving \~6% faster convergence.

---

### Official Review · Reviewer_rAmz · 2025-11-01

**Soundness:** 3
**Presentation:** 3
**Contribution:** 3
**Rating:** 6
**Confidence:** 3

**Summary:**

The paper introduces MuonBP, a variant of Muon that replaces per-step global gradient orthogonalization with block-local orthogonalization aligned to model-parallel shards, and performs a periodic full orthogonalization every P steps. This preserves Muon’s data efficiency while removing most cross-device communication on off-period steps. The authors (i) formalize MuonBP and its communication model, (ii) prove a convergence rate that interpolates between Muon and fully blocked Muon and shows two distinct step sizes (for block vs full steps) are theoretically preferable, and (iii) demonstrate up to 8% throughput gains at 8B scale with comparable or slightly better perplexity vs Muon, while BlockMuon is unstable or worse at scale.

**Strengths:**

The Originality, quality and signifance are good because of the following details:
1. Clear motivation with concrete bottleneck: Muon’s additional all-gathers cost 5–10% throughput vs AdamW under model parallelism, which MuonBP directly targets.
2. Simple, practical algorithmic knob: Period P trades fewer collectives for slighly weaker per-iteration conditioning and it aligns blocks to TP/FSDP shards to eliminate extra communications on off-period steps.
3. Compelling empirical evidence: At 8B, 8% throughput gain vs Muon with no loss in perplexity. BlockMuon underperforms/unstable at higher LRs, validating the need for periodic global steps.
4. Wall-clock wins: For fixed perplexity targets, MuonBP reaches them 10–13% faster, and for fixed time budgets it achieves 5–7% lower perplexity.

The clarity is great because its presentation is easy to follow. The figures are intuitive.

**Weaknesses:**

1. Limited exploration of P scheduling. The paper largely uses a fixed P (often 5). It will be better to add ablations with adaptive P and report wall-clock to target perplexity.

**Questions:**

Is there any rule of thumb regarding how to choose a good value of P?

---

> ### Author Response · Authors · 2025-11-20
>
> Thank you for your review and detailed evaluation of our work. We really appreciate your comments on the clarity, motivation, and significance of our work.
> - **On scheduling the period P.** We performed additional experiments on scheduling P. We use the same 960M parameters model architecture as in Section 4.2, and we run it for 2.37 billion tokens over 3000 iterations with batch size 96 and TP degree 8\. We found that a schedule that increases P _linearly_ from 2 to 20 over the course of training worked well. Our experimental results show that Muon (baseline) achieves 18.28 validation perplexity, while BlockMuon achieves 28.6 validation perplexity, and MuonBP (scheduled P: 2 \-\> 20\) achieves 18.01 final validation perplexity. Notably, MuonBP with this schedule reached Muon’s final perplexity of 18.28 around step 2800 (\~145 minutes) compared to the 154.3 minutes it takes Muon, achieving \~6% faster convergence to the same quality. MuonBP with a fixed P=11 (expected value of P with linear scheduling 2 \-\> 20\) achieved a slightly higher final validation perplexity of 18.11. As typically the differences between these algorithms are amplified with scale, this seems promising and we believe further exploration of scheduling P is an interesting avenue for future work.
> - **For a rule of thumb for choosing a fixed P**. We recommend running training for a short period and selecting the best-performing P in that duration, but this run should be with the same model size as the final run since model size strongly influences the throughput/throughput reduction compared to coordinate-wise optimizers. Our experiments show that the optimal P for short runs remains effective for longer. For example, P=5 performed well for both our 1.2B model trained on 14B tokens and when extended to 42B tokens. The optimal P will depend on the specific network topology and the throughput penalty of full orthogonalization. According to our theoretical analysis, it also depends on the ratio between the smoothness constant of the block norm and the operator norm.

---

### Official Review · Reviewer_nphQ · 2025-11-01

**Soundness:** 3
**Presentation:** 3
**Contribution:** 3
**Rating:** 6
**Confidence:** 3

**Summary:**

This paper addresses a key performance bottleneck in the Muon optimizer, which suffers from reduced throughput in model-parallel training due to the communication overhead of its gradient orthogonalization step. The authors propose Muon with Block-Periodic Orthogonalization (MuonBP), a variant that eliminates this overhead on most steps by performing orthogonalization independently on the local gradient shards on each device. To maintain the stability and convergence of standard Muon, a full, cross-device gradient orthogonalization is performed only periodically. The authors provide theoretical analysis justifying this periodic approach and the use of two distinct learning rates for block and full steps. Experiments on an 8B parameter model demonstrate that MuonBP matches Muon's performance while increasing throughput by 8%, effectively closing the throughput gap with coordinate-wise optimizers like AdamW.

**Strengths:**

1. The paper provides a solid theoretical analysis of the proposed MuonBP algorithm. The convergence guarantees (Theorem 2) insightfully connect the block-periodic approach to the two extremes (BlockMuon and standard Muon) and justify the novel use of two distinct learning rates, which adds rigor to the method.

2. The experimental validation is large-scale, convincing, and directly addresses the practical problem of throughput. By testing on models up to 8B parameters with 8-way tensor parallelism, the authors convincingly demonstrate that MuonBP achieves its goal of improving wall-clock throughput (by ~8%) without sacrificing the final model performance.

**Weaknesses:**

1. The core idea of using local updates to reduce communication is a well-studied direction in communication-efficient distributed training. While applying this idea to the Muon optimizer and achieving impressive experimental results is valuable, the concept itself is not entirely novel. A more fundamental solution to this problem would involve improving the all-gather operation itself to decouple the Newton-Schulz iteration from the communication, but this is admittedly a very difficult problem. Therefore, the paper's overall contribution is still commendable.

**Questions:**

1. I am curious about the key phenomenon in the experiments that explains why MuonBP is faster (in terms of wall-clock time, as seen in Figure 3) than standard Muon. Intuitively, MuonBP performs more local updates where the parameters are exposed to less global information (only the local shard gradient), which suggests it should not converge faster per step than standard Muon, which sees the full gradient. Could the authors elaborate on this surprising result?

---

> ### Author Response · Authors · 2025-11-20
>
> Thank you for your review. We appreciate your evaluation of Theorem 2 as insightful and your positive evaluation of our experimental results.
>
> - **On the connection to local updates.** We agree on intermittent communication, our method is very much inspired by algorithms such as Local SGD. That said, we would like to point out that in distributed optimization/learning algorithms like Local SGD, intermittency is used to aggregate updates before synchronization. In our work, we use it for orthogonalization instead and do not perform aggregation; it’s a very different operation, as iterated block orthogonalization does not correspond to global orthognalization. Instead, block orthogonalization corresponds to applying descent w.r.t a different norm, and that’s why the theoretical analysis is crucial to show that this idea makes sense when applied here.
> - **On why MuonBP is faster.** The main advantage of MuonBP is that each step is cheaper: it takes less time to do the local updates, and Table 3 shows that this, on average, results in MuonBP training at a higher average throughput than Muon. And unlike BlockMuon, the intermittent application of full orthogonalization negates the destabilizing effect of only local updates. Since each step is slightly faster and on average performs no worse than baseline Muon, we get faster overall wall clock convergence. As for per-step, it is indeed surprising that MuonBP converges faster even on a per-step basis compared to Muon (though not by much). Two points stand out here: (a) the coefficients for Newton-Schulz orthogonalization in vanilla Muon do not correspond to the most accurate orthogonalization of the gradient/momentum buffer [1], therefore more accurate orthogonalization does not necessarily mean better performance. And moreover (b) it is not clear apriori whether blocking is harmful to performance. For example, Muon is applied only to 2D tensors, which means that if we have an attention matrix of dimensions (N_heads, Hdim, dim) we have to reshape it into (N_heads\*Hdim, dim) first, and often all three attention matrices are fused together into one matrix of dimensions (3\*N_heads\*Hdim, dim). However, this is a heuristic choice: we could instead choose to orthogonalize them separately, or even orthogonalize the different head matrices separately. Depending on the type of tensor parallelism performed and the degree, blocking might correspond to orthogonalizing several heads together at once and it is possible that this is why it can perform better. We agree that this is an intriguing question and we hope to explore it further in future work.
>
> [1] Jordan et al., Muon: an optimizer for hidden layers in neural networks. https://kellerjordan.github.io/posts/muon/

---

### Meta-Review · Area_Chair_BpZE · 2026-01-03

**Summary:**

This work proposes Muon with Block-Periodic Orthogonalization, a communication-efficient variant of the Muon optimizer designed to alleviate the throughput bottleneck caused by per-step global gradient orthogonalization in model-parallel training. MuonBP performs block-local orthogonalization on most steps and periodically applies a full cross-device orthogonalization to preserve stability and convergence. The authors formalize the method and provide convergence analysis showing that MuonBP interpolates between fully local and fully global variants, motivating the use of distinct learning rates for block and full steps. Experiments on an 8B-parameter model demonstrate that MuonBP matches the performance of Muon achieving up to an 8% throughput improvement.

Overall, the reviewers raised a small number of concerns and requests for clarification, mainly related to the intuition behind the method and the reported results. These included limited exploration of scheduling strategies for block-periodic orthogonalization, experiments beyond pretraining, and questions regarding novelty and comparisons in terms of iteration count rather than wall-clock time.

**Reviewer Concerns:**

Most of the concerns were addressed by the authors in their rebuttal. In particular, the concerns regarding novelty were addressed through clearer explanations and stronger connections to the related literature, while the concern about scheduling was addressed by extending the experimental evaluation. The concerns raised by Reviewer Wrom appear to be based on misunderstandings, and I believe the authors clarified these points effectively in their response.

**Reviewer Scores:**

I think after a reasonable discussion and engagement, Reviewer Wrom would likely increase their score, while other reviewers would not change their scores and would expect them to maintain their already positive scores.

---

### Decision · Program_Chairs · 2026-01-26

Accept (Poster)